# Towards Nonlinearity: The *p*-Regularity Theory

**DOI:** 10.3390/e27050518

**Published:** 2025-05-12

**Authors:** Ewa Bednarczuk, Olga Brezhneva, Krzysztof Leśniewski, Agnieszka Prusińska, Alexey A. Tret’yakov

**Affiliations:** 1Department of CAD/CAM Systems Design and Computer-Aided Medicine, Faculty of Mathematics and Information Sciences, Warsaw University of Technology, 00-661 Warszawa, Poland; ewa.bednarczuk@pw.edu.pl; 2Department of Mathematics, Miami University, Oxford, OH 45056, USA; 3System Research Institute, Polish Academy of Sciences, 02-106 Warsaw, Poland; 4Faculty of Science, University of Siedlce, 08-110 Siedlce, Poland; aprus@uws.edu.pl (A.P.); alexey.tretiyakov@uws.edu.pl (A.A.T.); 5Dorodnicyn Computing Center, Federal Research Center “Computer Science and Control”, Russian Academy of Sciences, Moscow 119333, Russia

**Keywords:** nonlinear optimization, operator equations, implicit function theorem, nonlinear differential equations, degenerate problems, numerical methods, singularities, *p*-regularity

## Abstract

We present recent advances in the analysis of nonlinear problems involving singular (degenerate) operators. The results are obtained within the framework of *p*-regularity theory, which has been successfully developed over the past four decades. We illustrate the theory with applications to degenerate problems in various areas of mathematics, including optimization and differential equations. In particular, we address the problem of describing the tangent cone to the solution set of nonlinear equations in singular cases. The structure of *p*-factor operators is used to propose optimality conditions and to construct novel numerical methods for solving degenerate nonlinear equations and optimization problems. The numerical methods presented in this paper represent the first approaches targeting solutions to degenerate problems such as the Van der Pol differential equation, boundary-value problems with small parameters, and partial differential equations where Poincaré’s method of small parameters fails. Additionally, these methods may be extended to nonlinear degenerate dynamical systems and other related problems.

## 1. Introduction

Many fundamental results in nonlinear analysis and classical numerical methods in Banach spaces *X* and *Y* rely on the regularity of the mapping F:X→Y at a given point $\bar{x}\in X$. The regularity of a Fréchet differentiable mapping *F* is commonly understood as the surjectivity of its Fréchet derivative $F^{\prime }$. However, a growing number of applications in areas such as partial differential equations, control theory, and optimization require the development of special approaches to deal with nonregular problems.

We present the theory of *p*-regularity, which originated in the 1980s with the aim of providing constructive tools for the analysis of nonregular problems. To date, the theory of *p*-regularity has found successful applications in various contexts and different areas of mathematics, as discussed in numerous papers. This paper highlights the most distinguished applications of the theory of *p*-regularity, with the goal of reviewing important results and indicating potential and promising directions for its future development and applications.

The theory of *p*-regularity, also known as higher-order regularity theory, offers a framework for studying nonlinear problems in situations where regularity assumptions are not satisfied. It focuses on utilizing higher-order derivatives to analyze and understand the behavior of mappings having first-order derivatives that are not onto or lack regularity.

**Definition** **1**(cf. Definition 1.16 of [1])**.**
*Let F:X→Y be a continuously differentiable mapping from an open set U⊂X of a Banach space X into a Banach space Y. A vector $\bar{x}\in U$ is called a *regular *point of F if $F^{\prime }\left(\bar{x}\right)$ maps X onto the entire space Y, expressed as $Im\;F^{\prime }\left(\bar{x}\right)=Y.$ If $Im\;F^{\prime }\left(\bar{x}\right)\neq Y$, we refer to $\bar{x}$ as a singular (nonregular, irregular, degenerate) point of F.*

### 1.1. Recollection of the Fundamental Results in the Regular Case

Regularity is a common assumption in many fundamental results of real and functional analysis, such as the inverse function theorem and the Implicit Function Theorem (IFT). In this section, we revisit some of these results. The theorems presented in this section have their roots in the following classical result.

**Theorem** **1** (Banach open mapping principle [2], see also [3])**.**
*Let X and Y be Banach spaces. For any linear and bounded single-valued mapping A:X→Y, the following properties are equivalent:*
 *(a)* *A is surjective.* *(b)* *A is open.* *(c)* *There is a constant κ>0, such that for all y∈Y there exists x∈X with Ax=y and ∥x∥≤κ∥y∥.*The Banach open mapping principle can be extended to nonlinear mappings in various ways. One such result is stated in the next theorem. Let $B_{X}\left(x,t\right)$ denote the open ball in *X* with center at *x* and radius *t*.

**Theorem** **2** (Graves’ Theorem [4])**.**
*Let X and Y be Banach spaces, and let F:X→Y be a continuous function with F(0)=0. Let A be a linear operator from X onto Y, and let κ>0 be the corresponding constant from Theorem 1. Suppose that there exists a constant δ>0 with $\delta <\kappa^{-1}$ and ε>0, such that*(1)$$|F\left(x_{1}\right)-F\left(x_{2}\right)-A\left(x_{1}-x_{2}\right)\left|\leq \delta \right|x_{1}-x_{2}|$$*for all $x_{1},x_{2}\in B_{X}\left(0,\varepsilon \right)$. Then, the equation y=F(x) has a solution $x\in B_{X}\left(0,\varepsilon \right)$ whenever $\left|y\right|\leq \kappa^{-1}-\delta$.*

Note that the assumption of differentiability of *F* at 0 is not made, as the concept of a strictly differentiable function was introduced several years after the publication of Graves’ work [4]. Instead, the surjectivity of the operator *A* is used. The proof of Theorem 2, along with its reformulation in terms of a strictly differentiable function *F* and related discussions, can be found in [5]. For historical remarks, refer to [6].

Both the inverse function theorem and the Implicit Function Theorem can be deduced from Theorem 3 below; see Theorem 1.20 in Section 1.2 of [1] for details. We should also mention that certain variants of the inverse function theorem can be derived directly from the Implicit Function Theorem; see Section 4.2 for details.

To state Theorem 3, let us recall the definition of the Banach constant, denoted by C(A), for a bounded linear operator *A* between Banach spaces *X* and *Y*, as given in [1]:(2)$$C\left(A\right)=\sup\left\{r\geq 0|B_{Y}\left(0,r\right)\subset A\left(B_{X}\left(0,1\right)\right)\right\}=\inf\{∥y∥|y\notin A\left(B_{X}\left(0,1\right)\right)\}.$$

**Theorem** **3** (Lyusternik–Graves Theorem, [1])**.**
*Let X and Y be Banach spaces. Suppose that F:X→Y is strictly differentiable and regular at $\bar{x}\in X$. Then, for any positive $r<C(F^{\prime }\left(\bar{x}\right))$, there exists an ε>0 such that*$$B_{Y}\left(F\left(x\right),rt\right)\subset F\left(B_{X}\left(x,t\right)\right),$$*whenever $∥x-\bar{x}∥<\varepsilon$ and 0≤t<ε.*

For a thorough analysis of numerous consequences of Theorem 3, we refer the reader to the paper written by Dmitruk, Milyutin, and Osmolovskii [7], where the theorem is called the “generalized Lyusternik theorem”. In the monographs by Dontchev [3], Ioffe [1], and Dontchev and Rockafellar [8], Theorem 3 is also called the “Lyusternik-Graves theorem”. From a more general point of view, the theorem is treated by Dontchev and Frankowska in [9,10].

One of the consequences of Theorem 3 is the description of tangent vectors to the level set of a continuously differentiable mapping *F* at regular points (see Section 4.1 below).

### 1.2. Generalizations

Since the early 1970s, due to theoretical interests and an increasing number of involved economic and industrial applications, a vast body of literature has been devoted to relaxing the surjectivity assumption of the derivative in the fundamental results, some of which are given above, while maintaining as much of their conclusions as possible. It is beyond the scope of this paper to provide an exhaustive survey of the existing generalizations of the theorems stated above. For the purpose of this paper, we can distinguish between generalizations exploiting higher-order derivatives (see, e.g., Frankowska [11]) and generalizations that attempt to relax the surjectivity assumption of the derivative without referring to higher-order derivatives (see, e.g., Ekeland [12], Hamilton [13], Bednarczuk, Leśniewski, and Rutkowski [14]). The theory of *p*-regularity belongs to the first group of generalizations where higher-order derivatives are involved.

In this manuscript, we present the main concepts and results of the *p*-regularity theory, which has been developing successfully for the last forty years. One of the main goals of the theory of *p*-regularity is to replace the operator of the first derivative, which is not surjective, by a special mapping that is onto. Nonlinear mappings analyzed within the framework of *p*-regularity theory are those for which the derivatives up to the order p−1 are not surjective at a given point $\bar{x}$, where p≥2. The main concept of the *p*-regularity theory is the construction of the *p*-factor-operator, which is surjective at the point $\bar{x}$ (see Definition 4). The special definition and the property of surjectivity of the *p*-factor operator lead to generalizations of the fundamental results of analysis, including IFT and some classical numerical methods. The *p*-factor operator is defined in such a constructive way that it efficiently replaces the nonsurjective first derivative in a variety of situations. The structure of the *p*-factor-operator is used as a basis for analyzing nonregular problems and for constructing numerical methods for solving degenerate nonlinear equations and optimization problems. We discuss these generalizations in this paper.

There are many publications that focus on the case of p=2 and use a 2-factor operator in a variety of applications. In this work, we consider a more general case of p≥2 and do not make some additional assumptions introduced and required in the publications of other authors.

In the framework of metric spaces, the concept related to the problems discussed in the present paper and attempting to generalize the classical results given above is the concept of metric regularity; see, for example, [1,3]. For a function *f* acting between Banach spaces *X* and *Y* and being strictly differentiable at the given point $\bar{x}$, Corollary 5.3 in [3] complements Theorem 5.1 in [3]. It concludes that metric regularity at $\bar{x}$ for $f(\bar{x})$ is equivalent to surjectivity of the Fréchet derivative $Df(\bar{x})$ of *f* at $\bar{x}$. In the case when $X=Y=R^{n}$, this is the same as the nonsingularity of the Jacobian matrix $\nabla f(\bar{x})$.

The theory of *p*-regularity and the apparatus of *p*-factor operators make it possible to create new methods in computational mathematics to solve nonlinear problems of mathematical physics, such as the Van der Pol differential equation, boundary-value problems with a small parameter, Partial Differential Equations (PDEs) where Poincaré’s method of small parameters fails, nonlinear degenerate dynamical systems, and others. This is associated with the proposed fundamentally new design (after Newton) of a numerical method for solving essentially (degenerate) problems, which is described in this paper. Moreover, the proposed approach will allow us to construct a new type of difference scheme in computational mathematics for solving problems of nonlinear mathematical physics that are stable and converge quickly to a solution.

This pertains to the numerical solution of nonlinear equations such as the nonlinear heat equation, Burgers equation, Korteweg–de Vries equation, Navier–Stokes equation, etc. It also enables us to reorganize Numerical Analysis in a novel way, with solutions obtained being related to the mentioned problems. All of this is applicable to the emerging prospects for developing new technologies and designs in Computational Mathematics for solving problems and models related to Artificial Intelligence, optimization problems, dynamical systems, optimal control problems, etc. New opportunities are emerging for modeling and researching neural networks and creating new architectures for supercomputing.

It is important to highlight the development of fundamentally new and innovative methods in computational mathematics. The resulting schemes were far from any previous designs, emerging from years of research into the structure of degeneracy—specifically, the structure of degenerate mappings and solution sets of degenerate systems. Analyzing these structures requires methods that differ significantly from those used in the analysis of linear problems, leading to entirely new forms of mathematical objects, such as the *p*-factor-operator: $F^{\left(p\right)}\left(\bar{x}\right){\left[h\right]}^{p-1},$ where h∈X. In general form, this operator is formally introduced in Definition 4.

In degenerate problems, where the first derivative operator is not surjective, the *p*-factor-operator serves as its replacement. At the same time, this research has uncovered a previously hidden nonlinear world with unexpectedly rich diversity (cf. Theorem 4). The range of possible new methods is remarkably broad, yet it follows a structured framework that remains stable under small changes—similar to regularization techniques. These methods can also be adjusted based on the specific problem being solved.

Moreover, recent studies revealed that the so-called ill-posed problems and essentially nonlinear ones are locally equivalent. This finding suggests that many important problems, such as inverse problems, can be solved using the *p*-factor method or *p*-factor regularization. This represents a new and promising direction in both theoretical mathematics and practical applications.

### 1.3. Aims and Scope

The main focus of this work is on analyzing and solving nonlinear equations of the form(3)F(x)=0,
and optimization problems of the form(4)minf(x)subjecttoF(x)=0,
where f:X→R and F:X→Y are sufficiently smooth mappings, and *X* and *Y* are Banach spaces. Many interesting applied nonlinear problems can be written in one of these forms.

Nonlinear mappings *F* and problems of the form (Equation 3) and (Equation 4) can be divided into two classes, called regular (or nonsingular) and singular (or degenerate). The classification depends on the mapping *F*, which is either regular (that is, $F^{\prime }\left(\bar{x}\right):X\to Y$ is onto for a given $\bar{x}\in X$) or singular (that is, if $F^{\prime }(\bar{x}$) is not onto). Roughly speaking, regular mappings are those for which the Implicit Function Theorem arguments can be applied, and singular problems are those for which they cannot, at least, be directly applied.

The purpose of this paper is to give an overview of methods and tools of the *p*-regularity theory, and to show how they can be applied to analyze and develop methods for solving singular (irregular, degenerate) nonlinear equations and equality-constrained optimization problems. The development of the theory of *p*-regularity started in approximately 1983–1984, with the concept of *p*-regularity introduced by Tret’yakov in [15,16].

One of the main results of the theory of *p*-regularity is a detailed description of the structure of the zero set {x∈X,F(x)=0} of a nonregular nonlinear mapping F:X→Y. It is interesting to note that there have been several examples in the history of mathematics when fundamental results were obtained independently in the same general time period. One such example related to the theory of *p*-regularity concerns theorems about the structure of the zero sets of an irregular mapping satisfying a special higher-order regularity condition. The result that we are referring to was simultaneously obtained by Buchner, Marsden and Schecter [17] and Tretyakov [16]. The approaches proposed in [17] and in [16] are the same. The difference is in motivation and the context for the main result in both papers. In [17], the structure of the zero set around a point where the derivative is not surjective was studied in the context of bifurcation theory. Theorem 1.3 in [17] is referred to as a blowing-up result. In Fink and Rheinboldt [18], it was noted that Theorem 1.3 in [17] was a powerful generalization of the Morse Lemma, and some interesting counterexamples for a naive approach to the Morse Lemma were found. The same theorem derived by Tretyakov [16] is one of the main results for the *p*-regularity theory. The result led to various theoretical developments and applications of the theory to nonregular (or degenerate) problems in many areas of mathematics. We should note that the results and constructions introduced by Marsden and Tret’yakov are the same in the completely degenerate case.

This paper is organized as follows. We discuss essential nonlinearity and singular mappings in Section 2. We then recall the main concepts and definitions of the *p*-regularity theory in Section 3. We discuss some classical results of analysis and methods for solving nonlinear problems via the *p*-regularity theory in Section 4. In each subsection, we focus on singular problems that illustrate that the classical results are not necessarily satisfied in the nonregular case. We present generalizations of the same classical results, which were derived during the last forty years using the constructions and definitions of the *p*-regularity theory.

In this manuscript, we consider a variety of applications. We start Section 4 with the Lyusternik theorem in Section 4.1. The Lyusternik theorem plays an important role in the description of the solution sets of nonlinear equations and feasible sets of optimization problems in the regular case. However, the classical Lyusternik theorem might not hold if mapping *F* is singular at a given point $\bar{x}$. The first generalization of the classical Lyusternik theorem for *p*-regular mappings was derived and proved simultaneously in [15,17]. It can be applied to describe the zero set of a *p*-regular mapping. Representation Theorem and Morse Lemma are also presented in Section 4.1. We continue with the consideration of the Implicit Function Theorem in Section 4.2. Numerous books and papers, such as [13,19], discuss the classical Implicit Function Theorem. However, the classical version of the theorem is not applicable when a mapping F:X×Y→Z is not regular, meaning that $F_{y}^{\prime }\left(\bar{x},\bar{y}\right):Y\to Z$ is not onto for some $\left(\bar{x},\bar{y}\right)$, where the index *y* denotes the partial derivative with respect to the variable *y* (for a more detailed explanation of the notation, see the “General Notation” below). We present a generalization of the Implicit Function Theorem for nonregular mappings in Section 4.2. In Section 4.3, we cover the *p*-factor Newton’s method for solving nonlinear Equation (Equation 3) and finding critical points of an unconstrained optimization problem. Optimality conditions for equality-constrained optimization problems and Lagrange multiplier theorems for the regular and degenerate cases are considered in Section 4.4. The modified Lagrange function method for 2-regular problems is covered in Section 4.5. Singular problems of the calculus of variations and optimality conditions for *p*-regular problems of the calculus of variations are considered in Section 4.6. The existence of solutions to nonlinear equations in regular and degenerate cases is covered in Section 4.7. The second-order nonlinear ordinary differential equations with boundary conditions are presented in Section 4.8. Newton interpolation polynomials and the *p*-factor interpolation method are considered in Section 4.9. We make some concluding remarks in Section 5.

#### General Notation

Let L(X,Y) be the space of all continuous linear operators from *X* to *Y*, and for a given linear operator Λ∈L(X,Y), let us denote its kernel and image by KerΛ={x∈X∣Λx=0} and ImΛ={y∈Y∣y=Λxforsomex∈X}, respectively. Also, $\Lambda^{*}:Y^{*}\to X^{*}$ denotes the adjoint of Λ, where $X^{*}$ and $Y^{*}$ denote the dual spaces of *X* and *Y*, respectively.

Let *p* be a natural number and let A:X×X×…×X(with p copies of X)→Y be a continuous symmetric *p*-multilinear mapping. The *p*-form associated with A is the map $A{\left[x\right]}^{p}:X\to Y$ defined by$$A{\left[x\right]}^{p}=A\left(x,x,\ldots ,x\right),$$
for x∈X, where all instances of *x* in the expression A(x,x,…,x) are the same. Alternatively, we may simply view $A{\left[\cdot \right]}^{p}$ as a homogeneous polynomial Q:X→Y of degree *p* with $Q\left(\alpha x\right)=\alpha^{p}Q\left(x\right)$. Therefore, the space of continuous homogeneous polynomials Q:X→Y of degree *p* is denoted by $Q^{p}\left(X,\;Y\right)$.

If F:X→Y is of class $C^{2}$, then its derivative $F^{\prime }$ at a point $\bar{x}\in X$ is a continuous linear operator $F^{\prime }\left(\bar{x}\right)\in L\left(X,\;Y\right)$. The second derivative $F^{\prime \prime }\left(\bar{x}\right)$ is a bilinear operator on X×X and can be viewed as a mapping from *X* to L(X,Y). See ([20], Chapter VIII) for further details.

If F:X→Y is of class $C^{p}$, we denote by $F^{\left(p\right)}\left(\bar{x}\right):X^{p}\to Y$ the *p*th-order derivative of *F* at a given point $\bar{x}$. This is a symmetric *p*-multilinear map from $X^{p}$ to *Y*. The associated *p*-form, also called the *p*th–order mapping, is defined as$$F^{\left(p\right)}\left(\bar{x}\right){\left[h\right]}^{p}=F^{\left(p\right)}\left(\bar{x}\right)\left(h,h,\ldots ,h\right).$$In particular, for p=2, we have$$F^{\prime \prime }\left(\bar{x}\right){\left[h\right]}^{2}=F^{\prime \prime }\left(\bar{x}\right)\left(h,h\right).$$

Furthermore, for a given *p*-multilinear map, we introduce the following key notation for the *p*-kernel of the *p*th-order mapping:$${\mathrm{Ker}}^{p}F^{\left(p\right)}\left(\bar{x}\right)=\left\{h\in X\;|\;F^{\left(p\right)}\left(\bar{x}\right)\;{\left[h\right]}^{p}=0\;\right\}.$$Here, *h* represents elements of *X* that are repeatedly applied in the multilinear mapping. This set is also referred to as the locus of $F^{\left(p\right)}\left(\bar{x}\right)$.

When F:X×Y→Z is a continuously differentiable mapping, we use the notation $F_{y}^{\prime }\left(\bar{x},\bar{y}\right)$ to denote the partial derivative of *F* with respect to *y*. Specifically, for $(\bar{x},\bar{y})\in X\times Y$, the operator $F_{y}^{\prime }\left(\bar{x},\bar{y}\right)$ is defined as the Fréchet derivative of *F* with respect to *y*, satisfying:$$\lim_{h\to 0}\frac{F\left(\bar{x},\bar{y}+h\right)-F\left(\bar{x},\bar{y}\right)-F_{y}^{\prime }\left(\bar{x},\bar{y}\right)h}{∥h∥}=0,$$
where h∈Y and $F_{y}^{\prime }\left(\bar{x},\bar{y}\right):Y\to Z$ is a continuous linear operator.

## 2. Essential Nonlinearity and Singular Mappings

Let X,Y be Banach spaces. Let F:X→Y be a mapping in $C^{1}\left(W\right)$, where *W* is a neighborhood of a point $\bar{x}\in X$. According to Definition 1, a mapping *F* is called regular at $\bar{x}$, if(5)ImF′(x¯)=Y.

The following lemma on the local representation of a regular mapping holds.

**Lemma** **1** (Lemma 1, Section 1.3.3 of [21])**.**
*Let X and Y be Banach spaces, let W be a neighborhood of a point $\bar{x}\in X$, and let F:X→Y be of class $C^{1}\left(W\right)$. If F is regular at $\bar{x}$, then there exist a neighborhood U of 0, a neighborhood V⊂W, and a diffeomorphism φ:U→V, such that:*
 *1.* *$\varphi \left(0\right)=\bar{x}$,* *2.* *$F\left(\varphi \left(x\right)\right)=F\left(\bar{x}\right)+F^{\prime }\left(\bar{x}\right)x$ for all x∈U,* *3.* *$\varphi^{\prime }\left(0\right)=I_{X}$ (the identity mapping on X).*

Lemma 1 states that the diffeomorphism φ transforms *F* locally into an affine mapping:(6)$$F\left(\varphi \left(x\right)\right)=F\left(\bar{x}\right)+F^{\prime }\left(\bar{x}\right)x\;\;\;\mathrm{for}\;\mathrm{all}\;x\in U.$$In other words, φ provides a local reparametrization under which *F* takes an affine form in *U*. This result is also known as the local “trivialization theorem” (Theorem 1.26 of [1]).

If the regularity condition (Equation 5) is not satisfied, then, in general, *F* cannot be locally linearized because such a diffeomorphism φ does not exist.

There exist many mappings that do not admit local linearization. The concept of essentially nonlinear mappings, introduced in [22], provides a formal framework for describing such cases.

**Definition** **2.** 
*Let V be a neighborhood of a point $\bar{x}$ in X, and let U⊂X be a neighborhood of 0. A mapping F:V→Y, where $F\in C^{2}\left(V\right)$, is said to be essentially nonlinear at $\bar{x}$ if there exists a perturbation of the form*

$$\tilde{F}\left(\bar{x}+x\right)=F\left(\bar{x}+x\right)+\omega \left(x\right),\;where\;∥\omega \left(x\right)∥=o(∥x∥),$$

*such that there does not exist any nondegenerate transformation φ:U→V, $\varphi \in C^{1}\left(U\right)$, satisfying $\varphi \left(0\right)=\bar{x}$, $\varphi^{\prime }\left(0\right)=I_{X}$, and such that Equation (Equation 6) holds with φ and $\tilde{F}$.*


We say that ∥ω(x)∥=o(∥x∥) as x→0 if $\;\lim_{x\to 0}\frac{\omega (x)}{∥x∥}=0.$ For example, if ω:X→R, then $\omega \left(x\right)={∥x∥}^{2}$ is o(∥x∥).

**Definition** **3.** 
*We say that the mapping F:X→Y is singular (or degenerate) at $\bar{x}\in X$ if it fails to be regular; that is, if its derivative is not onto:*

(7)
ImF′(x¯)≠Y.



The following Theorem 4, which establishes the relationship between the two notions of essential nonlinearity and singularity, was derived as Theorem 2.3 in [22]. We provide its proof here to complete our development.

**Theorem** **4.** 
*Suppose V is a neighborhood of a given point $\bar{x}\in X,$ and X,Y are Banach spaces. Suppose F:V→Y is in $C^{2}$. If $F(\bar{x})=0$, then F is essentially nonlinear at the point $\bar{x}$ if and only if F is singular at $\bar{x}$.*


**Proof.** Suppose *F* is singular at the point $\bar{x}$ and $F(\bar{x})=0$. Since ImF′(x¯)≠Y, there exists a nonzero element ξ∈Y, such that(8)ξ∉ImF′(x¯).Thus, ξ∈Y∖ImF′(x¯). Since $F^{\prime }\left(\bar{x}\right)$ is linear, we can assume ∥ξ∥=1.Assume on the contrary that *F* is not essentially nonlinear at $\bar{x}$. Define the mapping $\tilde{F}:X\to Y$ by(9)$$\tilde{F}\left(\bar{x}+x\right)=F\left(\bar{x}\right)+F^{\prime }\left(\bar{x}\right)x+\xi {∥x∥}^{2},$$
where ξ∥x∥2∉ImF′(x¯).By Definition 2, with $\tilde{F}$ defined in (Equation 9), there exist a neighborhood U⊂X of 0 and a nondegenerate transformation φ:U→V, $\varphi \in C^{1}\left(U\right)$, such that $\varphi \left(0\right)=\bar{x}$, $\varphi^{\prime }\left(0\right)=I_{X}$, and (Equation 6) holds with φ and $\tilde{F}$:(10)$$\tilde{F}\left(\varphi \left(x\right)\right)=\tilde{F}\left(\bar{x}\right)+{\tilde{F}}^{\prime }\left(\bar{x}\right)x=F\left(\bar{x}\right)+F^{\prime }\left(\bar{x}\right)x$$
for all x∈U.Since $F(\bar{x})=0$ and F′(x¯)x∈ImF′(x¯), it follows from (Equation 10) that(11)F˜(φ(x))∈ImF′(x¯).However, using $F(\bar{x})=0$, $\varphi \left(0\right)=\bar{x}$, and $\varphi^{\prime }\left(0\right)=I_{X}$, we obtain(12)$$\begin{array}{cc} \tilde{F}\left(\varphi \left(x\right)\right) & =F(\bar{x}+\left(\varphi \left(x\right)-\bar{x}\right))\\ \; & =F\left(\bar{x}\right)+F^{\prime }\left(\bar{x}\right)\left(\varphi \left(x\right)-\bar{x}\right)+\xi {∥\varphi \left(x\right)-\bar{x}∥}^{2}\\ \; & =F^{\prime }\left(\bar{x}\right)\left(\varphi \left(x\right)-\bar{x}\right)+\xi {∥\varphi \left(0\right)+\varphi^{\prime }\left(0\right)x+\omega_{1}\left(x\right)-\bar{x}∥}^{2}\\ \; & =F^{\prime }\left(\bar{x}\right)\left(\varphi \left(x\right)-\bar{x}\right)+\xi {∥x+\omega_{1}\left(x\right)∥}^{2}, \end{array}$$
where $∥\omega_{1}\left(x\right)∥=o(∥x∥)$. Thus, for small *x*,$$\xi ∥x+\omega_{1}{\left(x\right)∥}^{2}\neq 0.$$Taking into account that ξ∥x∥2∉ImF′(x¯) for any x∈V, along with Equation (Equation 12) and the fact that F′(x¯)(φ(x)−x¯)∈ImF′(x¯), we conclude that(13)F˜(φ(x))∉ImF′(x¯).This contradicts (Equation 11), and therefore *F* is essentially nonlinear at $\bar{x}$.To prove the converse, suppose that *F* is essentially nonlinear at $\bar{x}$ but not singular; that is, suppose *F* is regular at this point.By the persistence of the regularity condition, for any perturbation$$\tilde{F}\left(\bar{x}+x\right)=F\left(\bar{x}+x\right)+\omega \left(x\right),$$
where ∥ω(x)∥=o(∥x∥), the map $\tilde{F}\left(\bar{x}+x\right)$ remains regular at $\bar{x}$, and $F^{\prime }\left(\bar{x}\right)={\tilde{F}}^{\prime }\left(\bar{x}\right)$. Hence, by Lemma 1, $\tilde{F}\left(\bar{x}+x\right)$ can be written as(14)$$\tilde{F}\left(\varphi \left(x\right)\right)=\tilde{F}\left(\bar{x}\right)+{\tilde{F}}^{\prime }\left(\bar{x}\right)x,$$
where $\varphi \left(0\right)=\bar{x}$ and $\varphi^{\prime }\left(0\right)=I_{X}$. This contradicts the definition of the essential nonlinearity of the mapping *F*. □

Under additional splitting assumptions, which are not made here, the representation (Equation 14) would be a standard consequence of the IFT, as in, for example, ([23], §2.5).

## 3. Elements of p-Regularity Theory

For the purpose of describing essentially nonlinear problems, a concept of *p*-regularity was introduced by Tret’yakov [15,16,24] using the notion of a *p*-factor operator. In this section, we introduce the main definitions of the *p*-regularity theory, as presented, for example, in [21,22,24].

Let *X* and *Y* be Banach spaces. Suppose that F:X→Y is a $C^{p}$-class mapping that is singular (nonregular) at a given point $\bar{x}\in X$. We construct the *p*-factor operator under the assumption that the space *Y* can be decomposed into the (topological) direct sum(15)$$Y=Y_{1}⊕\ldots ⊕Y_{p},$$
where Y1=cl(ImF′(x¯)), is the closure of the image of the first derivative of *F* evaluated at $\bar{x}$. To define the remaining spaces, let $S_{1}=Y$ and let $S_{2}\subset Y$ be a closed complementary subspace to $Y_{1}$, that is, $Y=Y_{1}⊕S_{2}$ if $S_{2}$ exists. Next, let $P_{S_{2}}:Y\to S_{2}$ be the projection operator onto $S_{2}$ along $Y_{1}$. Define $Y_{2}$ as the closed linear span of the projection of the quadratic map image:Y2=cl(spanImPS2F(2)(x¯)[h]2).More generally, define $Y_{i}$ inductively as follows:Yi=cl(spanImPSiF(i)(x¯)[h]i)⊆Si,i=2,…,p−1,p>2,
where $S_{i}$ is a choice of a closed complementary subspace for $Y_{1}⊕\ldots ⊕Y_{i-1}$ with respect to *Y*, and $P_{S_{i}}:Y\to S_{i}$ is the projection operator onto $S_{i}$ along $Y_{1}⊕\ldots ⊕Y_{i-1}$ for i=2,…,p. Finally, let $Y_{p}=S_{p}.$ The order *p* is the minimum number for which (Equation 15) holds. In particular, for p=2, we have $Y=S_{1}=Y_{1}⊕S_{2}$. When *Y* is a Hilbert space, there exists a complementary subspace to $Y_{1}$, namely the orthogonal subspace $Y_{2}=Y_{1}^{⊥}$.

**Remark** **1.** *The subspaces $Y_{i}$ in assumption *(Equation 15)* can be replaced, in further considerations, by subspaces constructed using the so-called factorization procedure. Specifically, we define*Y1=cl(ImF′(x¯)),*as before. However, instead of $Y_{2}$, we use the space $Y/Y_{1}$, called* the quotient (or factor) space. *Note that the quotient space is itself a Banach space (see, e.g., [25]). Moreover, if decomposition* (Equation 15) *holds, then $Y_{2}$ is isomorphic to $Y/Y_{1}$. For simplicity of presentation, we continue to use assumption* (Equation 15)*.*

Define the following mappings (see Tret’yakov [24]):(16)$$f_{i}:X\to Y_{i},\;f_{i}\left(x\right)=P_{Y_{i}}F\left(x\right),\;i=1,\;\ldots ,p,$$
where $P_{Y_{i}}:Y\to Y_{i}$ is the projection operator onto $Y_{i}$ along $Y_{1}⊕\ldots ⊕Y_{i-1}⊕Y_{i+1}⊕\ldots ⊕Y_{p}$ with respect to *Y* for i=1,…,p. Recall that $P_{Y_{i}}$ is the projection onto $Y_{i}$ along (or parallel to) $W_{i}=Y_{1}⊕\ldots ⊕Y_{i-1}⊕Y_{i+1}⊕\ldots ⊕Y_{p}$ if $\mathrm{Ker}P_{Y_{i}}=W_{i}$.

In our notation, $f_{i}^{\left(k\right)}\left(\bar{x}\right)$ denotes the *k*-th derivative of $f_{i}$ at $\bar{x}$. By the construction of the subspaces $Y_{i}$, we have(17)$$f_{i}^{\left(k\right)}\left(\bar{x}\right)=P_{Y_{i}}F^{\left(k\right)}\left(\bar{x}\right)=0,\;i=1,\ldots ,p,\;k=1,\ldots ,i-1.$$

We define a mapping *F* as completely degenerate up to order *p* if(18)$$F^{\left(k\right)}\left(\bar{x}\right)=0\;for\;k=1,\ldots ,p-1.$$

**Remark** **2.** *If the mapping F is* completely degenerate *up to order p, then* (Equation 17) *implies that each mapping $f_{i}$, defined in *(Equation 16)*, is also completely degenerate at $\bar{x}$ up to order i−1 for i=1,…,p. That is,*$$f_{i}^{\left(k\right)}\left(\bar{x}\right)=0,\;k=1,\ldots ,i-1,\;i=1,\ldots ,p.$$

With all the notation established above, we are now ready to define the *p-factor operator*.

**Definition** **4.** *For a fixed vector h∈X,h≠0, and mappings $f_{i}$, defined in* (Equation 16)*, the linear operator $\Psi_{p}\left(h\right)\in L\left(X,Y_{1}⊕\ldots ⊕Y_{p}\right)$,*
(19)$$\Psi_{p}\left(h\right)x=f_{1}^{\prime }\left(\bar{x}\right)x+f_{2}^{\prime \prime }\left(\bar{x}\right)\left[h\right]x+\ldots +f_{p}^{\left(p\right)}\left(\bar{x}\right){\left[h\right]}^{p-1}x,\;x\in X,$$*is called the p*-factor operator. *Alternatively, the following equivalent form can be used:*$$\Psi_{p}\left(h\right)x=f_{1}^{\prime }\left(\bar{x}\right)x+\frac{1}{2!}f_{2}^{\prime \prime }\left(\bar{x}\right)\left[h\right]x+\ldots +\frac{1}{p!}f_{p}^{\left(p\right)}\left(\bar{x}\right){\left[h\right]}^{p-1}x.$$

Note that when *F* is regular at $\bar{x}$, meaning ImF(x¯)=Y, we have $Y_{1}=Y$. In this case, the *p*-factor operator reduces to the operator of the first derivative: $\Psi_{1}\left(h\right)x=F^{\prime }\left(\bar{x}\right)x$ for any x∈X.

For p=2, the *p*-factor-operator (Equation 19) takes the form(20)$$\Psi_{2}\left(h\right)x=f_{1}^{\prime }\left(\bar{x}\right)x+f_{2}^{\prime \prime }\left(\bar{x}\right)\left[h\right]x,\;x\in X,$$
or, equivalently, $\Psi_{2}\left(h\right)x=f_{1}^{\prime }\left(\bar{x}\right)x+\frac{1}{2}f_{2}^{\prime \prime }\left(\bar{x}\right)\left[h\right]x$ for x∈X, where h∈X and h≠0. In view of (Equation 17), the construction of the operator $\Psi_{2}\left(h\right)$ (and $\Psi_{p}\left(h\right)$ in general) is closely tied to the decomposition of the image space (Equation 15). The idea is to use higher-order derivatives of *F* up to order *p* to obtain (Equation 15).

In particular, for p=2 and $\Psi_{2}\left(h\right)$ given by (Equation 20), we seek those h∈X that ensure the equality Imf2′′(x¯)[h](X)=S2, where $S_{2}$ is the complementary space to $Y_{1}$.

If a mapping *F* is completely degenerate up to order *p*, meaning that (Equation 18) holds, and ImF(p)(x¯)[h]p−1=Y, then the *p*-factor operator simplifies to $\Psi_{p}\left(h\right)x=F^{\left(p\right)}\left(\bar{x}\right){\left[h\right]}^{p-1}x$.

Recall that a bounded linear operator T:X→Y between Banach spaces *X* and *Y* is called Fredholm if the kernel of *T* has finite dimension and the image of *T* is a closed subspace of finite codimension in Y (see, for example ([26], Chapter 4) and [27]).

Hence, in the case of a Fredholm operator $F^{\prime }\left(\bar{x}\right)$, the subspace Y1=ImF′(x¯) has a complementary finite-dimensional subspace $Z_{2}$ such that $Y=Y_{1}⊕Z_{2}$.

With the *p*-factor operator $\Psi_{p}\left(h\right)$ defined in (Equation 19), we are now ready to state a few definitions of various types of *p*-regularity for a $C^{p}$-class mapping F:X→Y.

**Definition** **5.** *We say that the mapping F:X→Y is p*-regular at a given point $\bar{x}$ *along an element h∈X if*ImΨp(h)=Y.

**Remark** **3.** *The condition of p-regularity of the mapping F at the point $\bar{x}$ along h∈X is equivalent to the following condition:*(21)Imfp(p)(x¯)[h]p−1KerΨp−1(h)=Yp,*where $\Psi_{p-1}\left(h\right)=f_{1}^{\prime }\left(\bar{x}\right)+f_{2}^{\prime \prime }\left(\bar{x}\right)\left[h\right]+\ldots +f_{p-1}^{\left(p-1\right)}\left(\bar{x}\right){\left[h\right]}^{p-2}.$ In particular, when p=2, we have $\Psi_{1}\left(h\right)=f_{1}^{\prime }\left(\bar{x}\right)$, and condition *(Equation 21) *reduces to Imf2′′(x¯)[h]Kerf1′(x¯)=Y2, which follows from elementary algebra.*

We also define the *k*-kernel of the *k*th-order mapping $f_{k}^{\left(k\right)}\left(\bar{x}\right)$ as follows:(22)$${\mathrm{Ker}}^{k}f_{k}^{\left(k\right)}\left(\bar{x}\right)=\left\{h\in X|f_{k}^{\left(k\right)}\left(\bar{x}\right){\left[h\right]}^{k}=0\right\}.$$

**Definition** **6.** *We say the mapping F is p-regular at $\bar{x}$ if it is p-regular along any h from the set*(23)$$H_{p}\left(\bar{x}\right)=\left\{h\in X|h\in \overset{p}{\underset{i=1}{⋂}}{\mathrm{Ker}}^{i}f_{i}^{\left(i\right)}\left(\bar{x}\right)\right\}\setminus \left\{0\right\},$$*where the i-kernel of the ith-order mapping $f_{i}^{\left(i\right)}\left(\bar{x}\right)$ is defined in *(Equation 22).

For a linear surjective operator $\;\Psi_{p}\left(h\right):X\to Y$ between Banach spaces, we denote its right inverse by $\;{\left\{\Psi_{p}\left(h\right)\right\}}^{-1}$ (see [28]). Therefore, ${\left\{\Psi_{p}\left(h\right)\right\}}^{-1}:Y\to 2^{X}$ and we have(24)$${\left\{\Psi_{p}\left(h\right)\right\}}^{-1}\left(y\right)=\left\{x\in X|\Psi_{p}\left(h\right)x=y\right\}.$$We define the *norm* of ${\left\{\Psi_{p}\left(h\right)\right\}}^{-1}$ by(25)$$∥{\left\{\Psi_{p}\left(h\right)\right\}}^{-1}∥=\underset{∥y∥=1}{\sup}\inf\{∥x∥|x\in {\left\{\Psi_{p}\left(h\right)\right\}}^{-1}\left(y\right)\}.$$We say that ${\left\{\Psi_{p}\left(h\right)\right\}}^{-1}$ is *bounded* if $∥{\left\{\Psi_{p}\left(h\right)\right\}}^{-1}∥<\infty .$

**Definition** **7.** 
*A mapping $F\in C^{p}$ is called strongly p-regular at a point $\bar{x}$ if there exists α>0 such that*

$$\underset{h\in H_{p}^{\alpha }\left(\bar{x}\right)}{\sup}∥{\left\{\Psi_{p}\left(h\right)\right\}}^{-1}∥<\infty ,$$

*where ${\left\{\Psi_{p}\left(h\right)\right\}}^{-1}$ is the right inverse operator of $\Psi_{p}\left(h\right)$ and*

$$H_{p}^{\alpha }\left(\bar{x}\right)=\left\{h\in X\;|\;∥f_{i}^{\left(i\right)}\left(\bar{x}\right){\left[h\right]}^{i}∥\leq \alpha \;\mathrm{for}\mathrm{all}\;i=1,\ldots ,p,\;∥h∥=1\right\}.$$



The following examples illustrate the construction of the *p*-factor operator for the cases p=2 and p=3.

**Example** **1.** *Consider the mapping $F:R^{2}\to R^{2}$ defined by*$$F\left(x\right)=\left(\begin{array}{c} x_{1}+x_{2}\\ x_{1}x_{2} \end{array}\right).$$*Let $\bar{x}={\left(0,0\right)}^{T}$. Then, the Jacobian $F^{\prime }\left(\bar{x}\right)=\left(\begin{array}{cc} 1 & 1\\ 0 & 0 \end{array}\right)$ is singular (degenerate) at $\bar{x}.$ Hence, ImF′(x¯)=span{(1,0)}≠R2. Let $Y_{1}=\mathrm{span}\left\{\left(1,0\right)\right\}$ and $Y_{2}=\mathrm{span}\left\{\left(0,1\right)\right\}.$ To construct the 2-factor operator, we use the projection matrices*$$P_{Y_{1}}=\left(\begin{array}{cc} 1 & 0\\ 0 & 0 \end{array}\right)\;and\;P_{Y_{2}}=\left(\begin{array}{cc} 0 & 0\\ 0 & 1 \end{array}\right).$$*According to Equation *(Equation 16)*, the mappings $f_{1}:R^{2}\to Y_{1}$ and $f_{2}:R^{2}\to Y_{2}$ have the form*$$f_{1}\left(x\right)=\left(\begin{array}{c} x_{1}+x_{2}\\ 0 \end{array}\right)\;and\;f_{2}\left(x\right)=\left(\begin{array}{c} 0\\ x_{1}x_{2} \end{array}\right).$$*Then*$$f_{1}^{\prime }\left(x\right)=\left(\begin{array}{cc} 1 & 1\\ 0 & 0 \end{array}\right),\;f_{2}^{\prime }\left(x\right)=\left(\begin{array}{cc} 0 & 0\\ x_{2} & x_{1} \end{array}\right)$$*and*$$f_{2}^{\prime \prime }\left(x\right)h=\left(\begin{array}{cc} 0 & 0\\ h_{2} & h_{1} \end{array}\right).$$
*Hence, for $h={\left(h_{1},h_{2}\right)}^{T}\in R^{2}$, the 2-factor operator is defined by*

$$\Psi_{2}\left(h\right)=f_{1}^{\prime }\left(\bar{x}\right)+f_{2}^{\prime \prime }\left(\bar{x}\right)h=\left(\begin{array}{cc} 1 & 1\\ h_{2} & h_{1} \end{array}\right).$$

*It can be verified that the 2-factor operator is surjective whenever $h_{1}\neq h_{2}$.*

*In this example, we have*

$${\mathrm{Ker}}^{1}f_{1}^{\prime }\left(\bar{x}\right)=\mathrm{span}\left\{\left(1,-1\right)\right\}\;and\;{\mathrm{Ker}}^{2}f_{2}^{\prime \prime }\left(\bar{x}\right)=\mathrm{span}\left\{\left(1,0\right)\right\}\cup \mathrm{span}\left\{\left(0,1\right)\right\}.$$

*This result implies that $H_{2}\left(\bar{x}\right)=\emptyset$. Hence, according to Definition 5, the mapping F is 2-regular at $\bar{x}$ along any h∈X with $h_{1}\neq h_{2}$, but it is not 2-regular at $\bar{x}$. As we observe, it may happen that F is 2-regular along some h∈X but $H_{2}\left(\bar{x}\right)=\emptyset$. Therefore, a given mapping F may fail to be 2-regular with respect to all h∈X, h≠0.*


**Example** **2.** 
*Case p=3. Consider mapping $F:R^{2}\to R^{3}$ defined by*

$$F\left(x\right)=\left(\begin{array}{c} x_{1}+x_{2}\\ x_{1}x_{2}^{2}\\ x_{1}^{3} \end{array}\right).$$

*With $\bar{x}={\left(0,0\right)}^{T}=0$, we obtain*

$$F^{\prime }\left(x\right)=\left(\begin{array}{cc} 1 & 1\\ x_{2}^{2} & 2x_{1}x_{2}\\ 3x_{1}^{2} & 0 \end{array}\right)\;and\;F^{\prime }\left(0\right)=\left(\begin{array}{cc} 1 & 1\\ 0 & 0\\ 0 & 0 \end{array}\right).$$

*Then, with $h={\left(h_{1},h_{2}\right)}^{T}$,*

$$F^{\prime }\left(x\right)h=\left(\begin{array}{c} h_{1}+h_{2}\\ x_{2}^{2}h_{1}+2x_{1}x_{2}h_{2}\\ 3x_{1}^{2}h_{1} \end{array}\right),$$


$$F^{\prime \prime }\left(x\right)\left[h\right]h=F^{\prime \prime }\left(x\right){\left[h\right]}^{2}=\left(\begin{array}{c} 0\\ 4x_{2}h_{1}h_{2}+2x_{1}h_{2}^{2}\\ 6x_{1}h_{1}^{2} \end{array}\right),\;F^{\prime \prime \prime }\left(x\right){\left[h\right]}^{2}=\left(\begin{array}{cc} 0 & 0\\ 2h_{2}^{2} & 4h_{1}h_{2}\\ 6h_{1}^{2} & 0 \end{array}\right).$$

*In this example,*Y1=ImF′(0)=span{(1,0,0)},Y2=(0,0,0),Y3=span{(0,1,0),(0,0,1)}.*To construct the 3-factor operator, we use the projection matrices*$$P_{Y_{1}}=\left(\begin{array}{ccc} 1 & 0 & 0\\ 0 & 0 & 0\\ 0 & 0 & 0 \end{array}\right)\;and\;P_{Y_{3}}=\left(\begin{array}{ccc} 0 & 0 & 0\\ 0 & 1 & 0\\ 0 & 0 & 1 \end{array}\right).$$*Then, using Equation *(Equation 16)*, we define $f_{1}$ and $f_{3}$ as follows:*$$f_{1}\left(x\right)=P_{Y_{1}}F\left(x\right)=\left(\begin{array}{c} x_{1}+x_{2}\\ 0\\ 0 \end{array}\right)\;and\;f_{3}\left(x\right)=P_{Y_{3}}F\left(x\right)=\left(\begin{array}{c} 0\\ x_{1}x_{2}^{2}\\ x_{1}^{3} \end{array}\right).$$
*By the definition of the 3-factor-operator, we obtain*

$$\begin{array}{cc} \Psi_{3}\left(h\right) & =f_{1}^{\prime }\left(\bar{x}\right)+f_{3}^{\prime \prime \prime }\left(\bar{x}\right){\left[h\right]}^{2},\\ & =\left(\begin{array}{cc} 1 & 1\\ 0 & 0\\ 0 & 0 \end{array}\right)+\left(\begin{array}{cc} 0 & 0\\ 2h_{2}^{2} & 4h_{1}h_{2}\\ 6h_{1}^{2} & 0 \end{array}\right)=\left(\begin{array}{cc} 1 & 1\\ 2h_{2}^{2} & 4h_{1}h_{2}\\ 6h_{1}^{2} & 0. \end{array}\right). \end{array}$$


*For $h=(h_{1},0)$, the 3-factor operator takes the form*

$$\Psi_{3}\left(h\right)=\left(\begin{array}{cc} 1 & 1\\ 0 & 0\\ 6h_{1}^{2} & 0 \end{array}\right)$$

*and $cl\;Im\Psi_{3}\left(h\right)=span\left\{\left(1,0,0\right),\left(1,0,1\right)\right\}$.*

*For $h=(0,h_{2})$, the 3-factor operator takes the form*

$$\Psi_{3}\left(h\right)=\left(\begin{array}{cc} 1 & 1\\ 2h_{2}^{2} & 0\\ 0 & 0 \end{array}\right)$$

*and $cl\;Im\Psi_{3}\left(h\right)=span\left\{\left(1,1,0\right),\left(1,0,0\right)\right\}$.*
*Now, using *(Equation 22)*, we determine the elements $h=(h_{1},h_{2})$ in the kernels by solving the following equations:*$$f_{1}^{\prime }\left(\bar{x}\right)h=\left(\begin{array}{c} h_{1}+h_{2}\\ 0\\ 0 \end{array}\right)=\left(\begin{array}{c} 0\\ 0\\ 0 \end{array}\right),\;f_{3}^{\prime \prime \prime }\left(\bar{x}\right){\left[h\right]}^{3}=\left(\begin{array}{c} 0\\ 6h_{1}h_{2}^{2}\\ 6h_{1}^{3} \end{array}\right)=\left(\begin{array}{c} 0\\ 0\\ 0 \end{array}\right).$$*Thus, we obtain*$$\mathrm{Ker}f_{1}^{\prime }\left(0\right)=\left\{h=\left(h_{1},h_{2}\right)\;|\;h_{1}+h_{2}=0\right\}=span\left\{\left(1,-1\right)\right\},$$*and*$$\mathrm{Ker}f_{3}^{\prime \prime \prime }\left(0\right)=\left\{h=\left(h_{1},h_{2}\right)\;|\;6h_{1}h_{2}^{2}=0,\;6h_{1}^{3}=0\right\}=span\left\{\left(0,1\right)\right\}.\;\;$$*Finally, one can verify that*$$\mathrm{Ker}f_{2}^{\prime \prime }\left(0,0\right)=R^{2}.$$

## 4. Singular Problems and Classical Results via the p-Regularity Theory

### 4.1. Lyusternik Theorem and Description of Solution Sets

The Lyusternik theorem plays an important role in describing solution sets of nonlinear equations and feasible sets of optimization problems in the regular case. This theorem has practical applications across various fields. It is particularly important in the study of optimization and variational problems. By characterizing the tangent cone, the theorem provides valuable information about critical points and the behavior of solutions in their vicinity. In control theory, the Lyusternik theorem can be used to analyze the stability and controllability of nonlinear control systems. By examining the tangent cone, one can gain insights into system behavior near critical points and determine the conditions necessary for stability and controllability. The Lyusternik theorem is also useful in the development and analysis of optimization algorithms, such as gradient-based methods. By characterizing the tangent cone, the theorem helps in designing efficient algorithms and understanding their convergence properties.

These are just a few examples of the practical applications of the Lyusternik theorem. Its insights into the tangent cone are valuable in many areas, including optimization, control theory, partial differential equations, and geometry, providing a deeper understanding of the behavior of solutions and critical points in a variety of mathematical problems.

Consider a nonlinear mapping F:U→Y, where *U* is a neighborhood of a point $\bar{x}\in X$. We are interested in the description of the *set* $M(\bar{x})$:(26)$$M\left(\bar{x}\right)=\left\{x\in U|F\left(x\right)=F\left(\bar{x}\right)\right\}.$$This notation highlights the fact that we will focus our attention on $\bar{x}.$ It is useful to recall the following definition of tangent vectors and tangent cones (see, for instance, [29]).

**Definition** **8.** 
*Let M be a subset of a Banach space X. A vector h∈X is said to be tangent to the set M at a point $\bar{x}\in M$ if there exist ε>0 and a mapping r(t), r:[0,ε]→X, such that*

$$\bar{x}+th+r\left(t\right)\in M\;\forall t\in \left[0,\varepsilon \right],$$

*and*

$$\lim_{t\to 0}\frac{∥r(t)∥}{t}=0.$$



The set of vectors tangent to *M* at the point $\bar{x}$ is called the *tangent cone* to the set *M* at $\bar{x}$, and is denoted by $T_{1}M\left(\bar{x}\right)$.

#### 4.1.1. Lyusternik Theorem in the Regular Case

In the regular case, the Lyusternik theorem (see [30]) can be formulated as follows.

**Theorem** **5** (Lyusternik theorem)**.**
*Let X and Y be Banach spaces, and let U be a neighborhood of $\bar{x}$ in X. Assume that F:U→Y is Fréchet differentiable on U, and that its derivative $F^{\prime }:U\to L\left(X,\;Y\right)$ is continuous at $\bar{x}$. Suppose further that F is regular at $\bar{x}$.**Then, the tangent cone to the set $M(\bar{x})$ defined in *(Equation 26)* coincides with the kernel of the operator $F^{\prime }\left(\bar{x}\right)$:*(27)$$T_{1}M\left(\bar{x}\right)=\mathrm{Ker}F^{\prime }\left(\bar{x}\right).$$

If *F* is singular at $\bar{x}$, then in some problems we may have $T_{1}M\left(\bar{x}\right)\neq \mathrm{Ker}F^{\prime }\left(\bar{x}\right)$, as illustrated in the following example.

**Example** **3.** 
*Let $X=R^{2}$, and let $x=\left(x_{1},x_{2}\right)\in R^{2}$. Define the mapping $F:R^{2}\to R$ by*

$$F\left(x\right)=x_{1}^{2}-x_{2}^{2}+{o(∥x∥}^{2}).$$

*Then, the derivative of F is given by $F^{\prime }\left(x_{1},x_{2}\right)=\left[2x_{1},-2x_{2}\right].$ Evaluating at $\bar{x}=\left(0,0\right)$, we obtain F(0,0)=0, $F^{\prime }\left(0,0\right)=\left(0,0\right)$, and $\mathrm{Ker}F^{\prime }\left(0,0\right)=R^{2}$. Calculating*

$$T_{1}M\left(0,0\right)=\mathrm{span}\left\{\left(1,1\right)\right\}\cup \mathrm{span}\left\{\left(1,-1\right)\right\},$$

*we conclude that $T_{1}M\left(0,0\right)\neq \mathrm{Ker}F^{\prime }\left(0,0\right)$.*


**Example** **4.** 
*Let F:C([0,1])→C([0,1]) be defined as F(x(t))=x(t). Then, the set*

M={x(t)∈C([0,1])∣F(x(t))=F(0)=0}={0}

*consists only of the zero function. The derivative of F, given by $F^{\prime }\left(x\left(t\right)\right)=I$, is surjective, where I is the identity operator on C([0,1]). Moreover, we have $\mathrm{Ker}F^{\prime }\left(0\right)=\left\{0\right\}=T_{1}M\left(0\right).$*


**Example** **5.** 
*Let $M=\left\{x\left(t\right)\in C\left(\left[0,1\right]\right)|\int_{0}^{1}\sin x\left(t\right)dt=\frac{2}{\pi }\right\}$ and define $\bar{x}\left(t\right)=\pi t$. To calculate $T_{1}M\left(\bar{x}\right)$, it is enough to apply Lyusternik’s theorem with F:C([0,1])→R given by $F\left(x\left(t\right)\right)=\int_{0}^{1}\sin x\left(t\right)dt.$*

*Using the trigonometric addition formulas, we obtain*

$$\frac{F(x+h)-F(x)}{∥h∥}=\int_{0}^{1}\sin x\left(t\right)\frac{\cos h(t)-1}{∥h(t)∥}dt+\int_{0}^{1}\cos x\left(t\right)\frac{\sin h(t)}{∥h(t)∥}dt.$$

*The first term on the right-hand side approaches 0 as ∥h(t)∥→0. In the second term, we use the fact that $\frac{\sin h(t)}{∥h(t)∥}$ approaches 1 as ∥h(t)∥→0.*

*Therefore, the derivative of F at $\bar{x}$ is given by*

$$F^{\prime }\left(\bar{x}\left(t\right)\right)\left(x\left(t\right)\right)=\int_{0}^{1}x\left(t\right)\cos\bar{x}\left(t\right)dt,$$

*which is surjective onto R. By Theorem 5, we conclude that*

$$T_{1}M\left(\bar{x}\right)=\mathrm{Ker}F^{\prime }\left(\bar{x}\right)=\left\{x\left(t\right)\in C\left(\left[0,1\right]\right)|\int_{0}^{1}x\left(t\right)\cos\bar{x}\left(t\right)dt=0\right\}.$$



The problem of describing solution sets in more general settings (e.g., nonlinear systems of inequalities) is approached qualitatively using metric regularity [31,32] and geometric derivability [33].

#### 4.1.2. A Generalization of the Lyusternik Theorem

Consider the problem of describing the *set* $M(\bar{x})$ in the nonregular case. As demonstrated in Example 3, the classical Lyusternik theorem 5 may not hold when *F* is singular at $\bar{x}$, so that $T_{1}M\left(\bar{x}\right)\neq \mathrm{Ker}F^{\prime }\left(\bar{x}\right)$.

The first generalization of the classical Lyusternik theorem for *p*-regular mappings was independently derived and proved in [15,17]; see also [21]. This generalization can be used to describe the zero set of a *p*-regular mapping.

**Theorem** **6** (Generalized Lyusternik theorem, [15])**.**
*Let X and Y be Banach spaces, and let U be a neighborhood of a point $\bar{x}\in X$. Assume that F:U→Y is a p–times continuously Fréchet differentiable mapping on U. Assume also that F is p-regular at $\bar{x}$.**Then,*$$T_{1}M\left(\bar{x}\right)=H_{p}\left(\bar{x}\right),$$*where the set $H_{p}\left(\bar{x}\right)$ is defined in *(Equation 23)*.*

The problem of describing the tangent cone to the solution set $M(\bar{x})$ of a nonlinear Equation (Equation 3) with a singular mapping *F* has also been studied in other papers (see, for example, [16,34]).

**Example** **6.** 
*To illustrate the statement of Theorem 6, define mapping $F:R^{3}\to R^{2}$ by*

(28)
$$F\left(x\right)=\left(\begin{array}{c} x_{1}^{2}-x_{2}^{2}+x_{3}^{2}\\ x_{1}^{2}-x_{2}^{2}+x_{3}^{2}+x_{2}x_{3} \end{array}\right).$$

*Consider $\bar{x}={\left(0,0,0\right)}^{T}.$ A straightforward computation shows that $F^{\prime }\left(\bar{x}\right)=0$, and*

$$F^{\prime \prime }\left(\bar{x}\right)=\left(\begin{array}{c} \left(\begin{array}{ccc} 2 & 0 & 0\\ 0 & -2 & 0\\ 0 & 0 & 2 \end{array}\right)\\ \left(\begin{array}{ccc} 2 & 0 & 0\\ 0 & -2 & 1\\ 0 & 1 & 2 \end{array}\right) \end{array}\right).$$

*Also,*

$${\mathrm{Ker}}^{2}F^{\prime \prime }\left(\bar{x}\right)=\mathrm{span}\left(\left[\begin{array}{c} 1\\ -1\\ 0 \end{array}\right]\right)⋃\mathrm{span}\left(\left[\begin{array}{c} 1\\ 1\\ 0 \end{array}\right]\right).$$


*Let $h={\left(1,1,0\right)}^{T}$ (or $h={\left(1,-1,0\right)}^{T}$), then ImF′′(x¯)h=R2. Hence, the mapping F(x) is 2-regular at $\bar{x}=0$. Then the statement of Theorem 6 in this example reduces to*

$$T_{1}M\left(\bar{x}\right)=H_{2}\left(\bar{x}\right)={\mathrm{Ker}}^{2}F^{\prime \prime }\left(\bar{x}\right)$$

*or*

$$T_{1}M\left(\bar{x}\right)={\mathrm{Ker}}^{2}F^{\prime \prime }\left(\bar{x}\right)=\mathrm{span}\left(\left[\begin{array}{c} 1\\ -1\\ 0 \end{array}\right]\right)⋃\mathrm{span}\left(\left[\begin{array}{c} 1\\ 1\\ 0 \end{array}\right]\right).$$



The next theorem presents another version of Theorem 6, which was formulated in [24] (see also [17,21] for additional results along these lines). To state the result, we denote by dist(x,M), the *distance function* from a point x∈X to a set *M*:$$\mathrm{dist}\left(x,M\right)=\underset{y\in M}{\inf}∥x-y∥,\;x\in X.$$

**Theorem** **7.** 
*Let X and Y be Banach spaces, and let U be a neighborhood of a point $\bar{x}\in X$. Assume that F:X→Y is a p-times continuously Fréchet differentiable mapping in U. Assume also that F is strongly p-regular at $\bar{x}$. Then, there exist a neighborhood $U^{\prime }\subseteq U$ of $\bar{x}$, a mapping $\xi \to x\left(\xi \right):U^{\prime }\to X$, and constants $\delta_{1}>0$ and $\delta_{2}>0$, such that for all $\xi \in U^{\prime }$ the following holds:*

$$F\left(\xi +x\left(\xi \right)\right)=F\left(\bar{x}\right),$$


(29)
$$\mathrm{dist}\left(\xi ,M\left(\bar{x}\right)\right)\leq ∥x\left(\xi \right)∥\leq \delta_{1}\sum_{i=1}^{p}\frac{∥f_{i}\left(\xi \right)-f_{i}\left(\bar{x}\right)∥}{∥\xi -\bar{x}{∥}^{i-1}},$$

*where $f_{i}$ are given by (Equation 16), and*

$$\mathrm{dist}\left(\xi ,M\left(\bar{x}\right)\right)\leq ∥x\left(\xi \right)∥\leq \delta_{2}\sum_{i=1}^{p}{∥f_{i}\left(\xi \right)-f_{i}\left(\bar{x}\right)∥}^{1/i}.$$



For the proof, see [21].

#### 4.1.3. Representation Theorem

The Representation Theorem is used in nonlinear analysis and is relevant to the study of the local behavior and representation of a mapping *F* around a special point $\bar{x}$. It also guarantees the existence of certain auxiliary mappings that have desirable properties and relate to the given mapping *F* and its local representation in some neighborhood of $\bar{x}$.

The Representation Theorem can be used, for example, in the study of optimization problems, particularly in constrained optimization. It helps in establishing the existence of critical points and characterizing their properties, which is essential for finding optimal solutions. The theorem is also relevant to variational methods, partial differential equations, and other areas of mathematical analysis. Moreover, it is useful in various numerical methods and computational techniques for approximating solutions of equations. Its versatility and utility stem from its ability to provide insights into the local behavior and representations of mappings near critical points, with wide-ranging applications in mathematical analysis and optimization. Its versatility and usefulness stem from its ability to provide insights into the local behavior and representations of mappings around critical points, which has wide-ranging applications in mathematical analysis and optimization.

To simplify the presentation of the next result, we state it for the case of the completely degenerate mapping *F*, defined in (Equation 18). Recall that in this case, ImF(p)(x¯)[h]p−1=Y, and the *p*-factor operator can be simplified to $\Psi_{p}\left(h\right)x=\frac{1}{p!}F^{\left(p\right)}\left(\bar{x}\right){\left[h\right]}^{p-1}x$.

**Theorem** **8** ([22])**.**
*Let X and Y be Banach spaces, and let V be a neighborhood of $\bar{x}$ in X. Suppose that F:V→Y is of class $C^{p+1}$, and that $F^{\left(i\right)}\left(\bar{x}\right)=0$ for i=1,…,p−1. Also assume the existence of a constant C>0, such that*$$\underset{∥h∥=1}{\sup}∥{\left\{F^{\left(p\right)}\left(\bar{x}\right){\left[h\right]}^{p-1}\right\}}^{-1}∥\leq C.$$*Then, there exist a neighborhood U of 0 in X, a neighborhood V of $\bar{x}$ in X, and mappings φ:U→X and ψ:V→X, such that φ and ψ are Fréchet-differentiable at 0 and $\bar{x}$, respectively, and the following hold:*
 *1.* *$\varphi \left(0\right)=\bar{x},$$\psi (\bar{x})=0$;* *2.* *$F\left(\varphi \left(x\right)\right)=F\left(\bar{x}\right)+\frac{1}{p!}F^{\left(p\right)}\left(\bar{x}\right){\left[x\right]}^{p}$ for all x∈U;* *3.* *$F\left(x\right)=F\left(\bar{x}\right)+\frac{1}{p!}F^{\left(p\right)}\left(\bar{x}\right){\left[\psi \left(x\right)\right]}^{p}$ for all x∈V;* *4.* *$\varphi^{\prime }\left(0\right)=\psi^{\prime }\left(\bar{x}\right)=I_{X}$.*

All assumptions of Theorem 8 are satisfied, for example, by the mapping$$F\left(x_{1},x_{2}\right)=x_{1}^{p}-x_{2}^{p}+x_{1}^{p+1}+x_{2}^{p+1},$$
where p≥2, p∈N. See also [35] for additional work on the representation theorem.

#### 4.1.4. Morse Lemma

The Morse Lemma is another fundamental result in analysis that relates the behavior of a smooth function near a nondegenerate critical point $\bar{x}$ to the local structure of its level sets. The Morse Lemma has several important applications in various areas of mathematics.

The Morse Lemma is used in differential geometry to analyze the behavior of geodesics and study the geometry of manifolds. By considering a function that measures the length or energy of curves on a manifold, the Morse Lemma allows us to understand the critical points of this function and their geometric implications. It provides insights into the existence, stability, and bifurcations of geodesics on a manifold.

The Morse Lemma has important applications in optimization and control theory, where it is used to analyze the behavior of objective functions and control systems near critical points. It helps characterize the local behavior of optimal solutions and understand stability properties. The Lemma can be employed to find critical points, perform sensitivity analysis, and study bifurcations in optimization problems and dynamical systems.

The Morse Lemma is also utilized in singularity theory, which focuses on the properties and classification of singular points or critical points of differentiable mappings. It provides a framework for understanding the local behavior of singularities and the ways in which their structure may change under small perturbations. The Lemma plays a key role in the classification and analysis of singular points and their stability.

The most interesting formulation of the Morse Lemma in the finite-dimensional case is given in the following lemma.

**Lemma** **2** (Morse Lemma)**.**
*Let $\bar{x}\in R^{n}$, and let $f:R^{n}\to R$ be a function of class $C^{3}\left(R^{n}\right)$, such that $f^{\prime }\left(\bar{x}\right)=0$ and the Hessian $f^{\prime \prime }\left(\bar{x}\right)$ is not degenerate. Then, in a neighborhood V of $\bar{x}$, there exist a curvelinear coordinate system $\left(y_{1},\ldots ,y_{n}\right)$ and an integer number k∈{0,…,n}, such that*$$f\left(x\right)=f\left(\bar{x}\right)+\sum_{i=1}^{k}y_{i}^{2}-\sum_{i=k+1}^{n}y_{i}^{2}$$*for all x∈V.*

**Proof.** Without loss of generality, we can assume that Hessian matrix $f^{\prime \prime }\left(\bar{x}\right)$ is diagonal:$$f^{\prime \prime }\left(\bar{x}\right)=\left(\begin{array}{ccccc} 1 & 0 & \ldots & 0 & 0\\ 0 & 1 & \ldots & 0 & 0\\ \vdots & \vdots & \ddots & \vdots & \vdots \\ 0 & 0 & \ldots & -1 & 0\\ 0 & 0 & \ldots & 0 & -1 \end{array}\right),$$
where for some number $k^{\prime }$ between 0 and *n*, the first $k^{\prime }$ columns have 1 on the main diagonal, and the other columns have −1. Otherwise, changing the basis, we can transform the Hessian to be a diagonal matrix.Then, in this case,$$f\left(x\right)=f\left(\bar{x}\right)+\sum_{i=1}^{k^{\prime }}{\left(x_{i}-x_{i}^{*}\right)}^{2}-\sum_{i=k^{\prime }+1}^{n}{\left(x_{i}-x_{i}^{*}\right)}^{2}+o(∥x-\bar{x}{∥}^{3}).$$Note that if the assumptions of the Morse Lemma hold, then the assumptions of the representation Theorem 8 are satisfied with p=2 and F=f. Hence, there exists a mapping ψ(x):U→R, such that$$f\left(\psi \left(x\right)\right)=f\left(\bar{x}\right)+\frac{1}{2}f^{\prime \prime }\left(\bar{x}\right){\left[x\right]}^{2},$$
where $∥\psi \left(x\right)-\left(x-\bar{x}\right)∥=o(∥x-\bar{x}∥)$ and $\psi^{\prime }\left(\bar{x}\right)=I_{X}$. It follows that $k=k^{\prime }$. Note that if $k=k^{\prime }$, then $\psi \left(x\right)=x-\bar{x}+o\left(x-\bar{x}\right)$, and if $k\neq k^{\prime }$, then we obtain a contradiction. Now, we can apply the statement of the representation Theorem 8 to the mapping y=ψ(x) to get the statement of Morse Lemma 2. □

See additional work on Morse Lemma in [36].

### 4.2. Implicit Function Theorem

In this section, we consider the equation F(x,y)=0, where F:X×Y→Z and *X*, *Y*, and *Z* are Banach spaces. Let $\left(\bar{x},\bar{y}\right)$ be a given point in X×Y that satisfies $F(\bar{x},\bar{y})=0$. We are interested in the existence of a mapping φ(x) defined in a neighborhood $U(\bar{x})$, such that $\varphi \left(x\right):U\left(\bar{x}\right)\to Y$ is a solution of the equation F(x,y)=0 near the given point $\left(\bar{x},\bar{y}\right)$. This mapping should satisfy the following conditions:(30)$$F\left(x,\varphi \left(x\right)\right)=0\;and\;\bar{y}=\varphi \left(\bar{x}\right).$$

#### 4.2.1. Implicit Function Theorem in the Regular Case

In the case when F:X×Y→Z is a continuously differentiable mapping, we denote its (Fréchet) derivative with respect to *y* at a point $(\bar{x},\bar{y})\in X\times Y$ by $F_{y}^{\prime }\left(\bar{x},\bar{y}\right):Y\to Z$.

In the case when *F* is regular at a point $\left(\bar{x},\bar{y}\right)$, meaning $F_{y}^{\prime }\left(\bar{x},\bar{y}\right)$ is onto, the classical Implicit Function Theorem (IFT) guarantees the existence of a smooth mapping φ(x) defined in a neighborhood $U(\bar{x})$, such that (Equation 30) holds and $∥\varphi \left(x\right)-\bar{y}∥\leq C∥F\left(x,\bar{y}\right)∥$ for all *x* in $U(\bar{x})$, where C>0. There are numerous books and papers devoted to the IFT, including [13,19]. Various formulations of the standard IFT exist, and Theorem 9 presents one such statement.

**Theorem** **9** (Implicit Function Theorem)**.**
*Let X and Y be Banach Spaces. Assume that F:X×Y→Z is continuously Fréchet differentiable at $(\bar{x},\bar{y})\in X\times Y$, $F(\bar{x},\bar{y})=0$, and that ImFy′(x¯,y¯)=Z. Then, there exist constants $C,C_{1}>0$, a sufficiently small δ>0, and a function $\varphi :B(\bar{x},\delta )\to Y$ such that, for $x\in B(\bar{x},\delta )$, the following holds:*(31)$$\varphi \left(\bar{x}\right)=\bar{y},\;F\left(x,\varphi \left(x\right)\right)=0,\;∥\varphi \left(x\right)-\bar{y}∥\leq C_{1}∥F\left(x,\bar{y}\right)∥\leq C∥x-\bar{x}∥.$$

The situation changes when the mapping F is degenerate (nonregular) at $\left(\bar{x},\bar{y}\right)$; that is, when $F_{y}^{\prime }\left(\bar{x},\bar{y}\right)$ is not onto. In this case, the classical IFT cannot be applied to guarantee the (local) existence of a solution y=φ(x). The importance of examining this situation arises from the need to solve various nonlinear problems, many of which, as shown in [22], are singular (degenerate).

#### 4.2.2. Implicit Function Theorem in the Degenerate Case

In this section, we focus on the case when mapping F:X×Y→Z is not regular; that is, when $F_{y}^{\prime }\left(\bar{x},\bar{y}\right)$ is not onto.

As an example, consider mapping F:R×R→R, $F\left(x,y\right)=x-y^{p}$, where p=2k+1 with some k∈N. If $\left(\bar{x},\bar{y}\right)=\left(0,0\right)$, then $F(\bar{x},\bar{y})=0$ and $F_{y}^{\prime }\left(\bar{x},\bar{y}\right)=0$, so the mapping *F* is not surjective. The classical IFT is not applicable in this case. However, there exists mapping $\varphi \left(x\right)=x^{1/p}$, such that $F\left(x,x^{1/p}\right)=x-{\left(x^{1/p}\right)}^{p}=0$. Moreover, $∥\varphi \left(x\right)-\bar{y}∥=∥F\left(x,\bar{y}\right){∥}^{1/p}$, and, by (Equation 31), the following inequality holds with C=1>0:$$∥\varphi \left(x\right)-\bar{y}∥\leq C∥F\left(x,\bar{y}\right){∥}^{1/p}.$$Thus, while the conditions of a standard implicit function are not satisfied in the example, the statement similar to (Equation 31) holds. The example serves as a motivation and illustration for the *p*-order IFT. To our knowledge, the first generalization of the IFT for nonregular mappings was formulated in [24]. Generalizations of the IFT for 2-regular mappings were obtained in [21,36]. We will present a few versions of the IFT for *p*-regular mappings in this section.

To simplify the presentation, we begin with Theorem 10, which is stated in Euclidean spaces. A slight modification of this theorem was derived in [21]. To formulate the theorem, we first need to define the operator $\Psi_{p}\left(h\right)$ related to the mapping F:X×Y→Z. To do so, and similarly to the mappings introduced in Section 3, we define the following mappings (see [24]):(32)$$f_{i}\left(x,y\right):X\times Y\to Z_{i},\;f_{i}\left(x,y\right)=P_{Z_{i}}F\left(x,y\right),\;i=1,\;\ldots ,p,$$
where $P_{Z_{i}}:Z\to Z_{i}$ is the projection operator onto $Z_{i}$ along $Z_{1}⊕\ldots ⊕Z_{i-1}⊕Z_{i+1}⊕\ldots ⊕Z_{p}$ with respect to *Z* for i=1,…,p. The definition of $Z_{i}$ is similar to the definition of the subspaces $Y_{i}$ in Section 3.

Now we are ready to present the definition of the linear operator $\Psi_{p}\left(h\right):Y\to Z_{1}\times \ldots \times Z_{p}$, which is similar to the operator $\Psi_{p}\left(h\right)$ defined in (Equation 19). Since the construction of the *p* factor-operators are similar, we retain the same notation to keep the presentation clear and consistent. For a fixed vector h∈Y, h≠0, and mappings $f_{i}$ defined in (Equation 16), the linear operator $\Psi_{p}\left(h\right)\in L\left(Y,Z_{1}⊕\ldots ⊕Z_{p}\right)$ is given by(33)$$\Psi_{p}\left(h\right)y={f_{1}^{\prime }}_{y}\left(\bar{x},\bar{y}\right)y+{f_{2}^{\prime \prime }}_{yy}\left(\bar{x},\bar{y}\right)\left[h\right]y+\ldots +{f_{p}^{\left(p\right)}}_{y\ldots y}\left(\bar{x},\bar{y}\right){\left[h\right]}^{p-1}y,\;y\in Y,$$
where y…y indicates that all derivatives are taken with respect to the same variable *y*, which belongs to *Y*.

Before stating Theorem 10, we introduce some additional notation that will be used:In the expression ${f_{i}^{\left(r\right)}}_{\underset{q}{\underset{︸}{x\ldots x}}\underset{r-q}{\underset{︸}{y\ldots y}}}\left(\bar{x},\bar{y}\right)$, *r* represents the total order of differentiation, where differentiation is performed *q* times with respect to *x* and r−q times with respect to *y*.While the notation ${\left[h\right]}^{r-1}$ appears in the definition (Equation 33) of the linear operator $\Psi_{p}\left(h\right)$, the expression ${f_{i}^{\left(r\right)}}_{\underset{q}{\underset{︸}{x\ldots x}}\underset{r-q}{\underset{︸}{y\ldots y}}}\left(\bar{x},\bar{y}\right)=0$ signifies that all components of the derivative are equal to zero.The subscript notation x…x (*q*-times) indicates partial differentiation with respect to the first variable *x* performed *q* times.For r=0, the notation $f_{i}^{\left(0\right)}\left(\bar{x},\bar{y}\right)$ represents the function value $f_{i}\left(\bar{x},\bar{y}\right)$ itself.

**Theorem** **10** (Implicit Function Theorem [22])**.**
*Suppose that X, Y, and Z are Euclidean spaces, and let W be a neighborhood of a point $\left(\bar{x},\bar{y}\right)$ in X×Y. Assume that F:W→Z is of class $C^{2}$. Suppose $F(\bar{x},\bar{y})=0$ and there exists a neighborhood $U(\bar{x})$ in X, such that the following conditions hold:*
*The Singularity Condition:*$${f_{i}^{\left(r\right)}}_{\underset{q}{\underset{︸}{x\ldots x}}\underset{r-q}{\underset{︸}{y\ldots y}}}\left(\bar{x},\bar{y}\right)=0,\;r=0,\ldots ,i-1,\;q=0,\ldots ,r-1,\;i=1,\ldots ,p;$$$${f_{i}^{\left(i\right)}}_{\underset{q}{\underset{︸}{x\ldots x}}\underset{i-q}{\underset{︸}{y\ldots y}}}\left(\bar{x},\bar{y}\right)=0,\;q=1,\ldots ,i-1,\;\;i=1,\ldots ,p.$$*The pth Order Regularity Condition at the Point $\left(\bar{x},\bar{y}\right)$:**The operator $\Psi_{p}\left(h\right)$ defined in *(Equation 33)* satisfies*$$\Psi_{p}\left(h\right)Y=Z$$*for all $h\in {\left\{{\left(\Psi_{p}\left(h\right)\right)}_{y}\right\}}^{-1}\left(-F\left(x,\bar{y}\right)\right)$ and all $x\in U(\bar{x})$, such that $F(x,\bar{y})\neq 0$.**The Banach Condition:**There exists a constant c>0 such that, for any z∈Z with ∥z∥=1, the following holds:*$$\Psi_{p}\left(h\right)y=z,\;∥y∥\leq c.$$*The Elliptic Condition with respect to x:**There exists a constant m>0 such that*$$∥f_{i}\left(x,\bar{y}\right)∥\geq m∥x-\bar{x}∥$$*for all $x\in U(\bar{x})$ and for all i=1,…,p.*
*If conditions 1 to 4 are satisfied, then for any ε>0, there exist δ>0 and K>0 such that $B\left(\bar{x},\delta \right)\subset U\left(\bar{x}\right)$, and there is a map $\varphi :B\left(\bar{x},\delta \right)\to B\left(\bar{y},\varepsilon \right)$ satisfying:*
 *(a)* 
*$\varphi \left(\bar{x}\right)=\bar{y}$;*
 *(b)* 
*F(x,φ(x))=0 for all $x\in B(\bar{x},\delta )$;*
 *(c)* 
*$∥\varphi \left(x\right)-\bar{y}∥\leq K\sum_{i=1}^{p}{∥f_{i}\left(x,\bar{y}\right)∥}^{1/i}$ for all $x\in B(\bar{x},\delta )$.*



The alternative version of the IFT for nonregular mappings, presented as Theorem 11, was proved in [37]. Before stating the theorem, we introduce the following definition (Definition 2.3 in [38]).

**Definition** **9.** 
*The mapping F:X×Y→Z is called uniformly p-regular over a set M in Y if*

$$\underset{h\in M}{\sup}∥{\left\{\Psi_{p}\left(\bar{h}\right)\right\}}^{-1}∥<\infty ,\;\bar{h}=\frac{h}{∥h∥},\;h\neq 0,$$

*where*

$$∥{\left\{\Psi_{p}\left(\bar{h}\right)\right\}}^{-1}∥=\underset{∥z∥=1}{\sup}\inf\{∥y∥\;\left|\;\Psi_{p}\left(h\right)\left[y\right]=z\right.\}.$$



Additionally, we define the mapping $\Phi_{p}:Y\to Z_{1}⊕\ldots ⊕Z_{p}$ by$$\Phi_{p}=\left(f_{1\;y}^{\prime }\left(\bar{x},\bar{y}\right),\frac{1}{2}f_{2\;yy}^{\prime \prime }\left(\bar{x},\bar{y}\right),\ldots ,\frac{1}{p!}f_{p\;y\ldots y}^{\left(p\right)}\left(\bar{x},\bar{y}\right)\right),$$
where$$\Phi_{p}{\left[y\right]}^{p}=\left(f_{1\;y}^{\prime }\left(\bar{x},\bar{y}\right)\left[y\right],\frac{1}{2}f_{2\;yy}^{\prime \prime }\left(\bar{x},\bar{y}\right){\left[y\right]}^{2},\ldots ,\frac{1}{p!}f_{p\;y\ldots y}^{\left(p\right)}\left(\bar{x},\bar{y}\right){\left[y\right]}^{p}\right).$$Under the assumption that $Z_{1}⊕\ldots ⊕Z_{p}=Z$, we also introduce the corresponding inverse multivalued operator $\Phi_{p}^{-1}$:$$\Phi_{p}^{-1}\left(z\right)=\left\{\eta \in Y\;\left|\;\left(f_{1y}^{\prime }\left(\bar{x},\bar{y}\right)\left[\eta \right],\ldots ,\frac{1}{p!}f_{p\;y\ldots y}^{\left(p\right)}\left(\bar{x},\bar{y}\right){\left[\eta \right]}^{p}\right)\right.=\left(z_{1},z_{2},\ldots ,z_{p}\right)\right\},$$
where $z_{i}\in Z_{i}$, i=1,…,p.

**Theorem** **11** (The *p*th-order IFT)**.**
*Let X, Y and Z be Banach spaces, and let $U(\bar{x})$ and $U(\bar{y})$ be sufficiently small neighborhoods of $\bar{x}\in X$ and $\bar{y}\in Y$, respectively. Suppose that $F\in C^{p+1}\left(X\times Y\right)$ and $F(\bar{x},\bar{y})=0$. Assume that the mappings $f_{i}\left(x,y\right)$, i=1,…,p, introduced in Equation *(Equation 32)*, satisfy the following conditions:*
(1)*The Singularity Condition:*fi(x…x︸q(y…y︸r−q(r)(x¯,y¯)=0,r=1,…,i−1,q=0,…,r−1,i=1,…,p,fi(x…x︸q(y…y︸i−q(i)(x¯,y¯)=0,q=1,…,i−1,i=1,…,p.(2)*The* p-*Factor Approximation Condition:**There exists a sufficiently small ε>0 such that, for all $y_{1},y_{2}\in \left(U\left(\bar{y}\right)\setminus \left\{\bar{y}\right\}\right)$, the following holds:*$$\begin{array}{cc} \; & ∥f_{i}\left(x,\bar{y}+y_{1}\right)-f_{i}\left(x,\bar{y}+y_{2}\right)-\frac{1}{i!}f_{i\;y\ldots y}^{\left(i\right)}\left(\bar{x},\bar{y}\right){\left[y_{1}\right]}^{i}+\frac{1}{i!}f_{i\;y\ldots y}^{\left(i\right)}\left(\bar{x},\bar{y}\right){\left[y_{2}\right]}^{i}∥\\ \; & \;\leq \varepsilon \left(∥y_{1}{∥}^{i-1}+{∥y_{2}∥}^{i-1}\right)∥y_{1}-y_{2}∥,\;i=1,\ldots ,p. \end{array}$$(3)*The Banach Condition:**There exists a nonempty open set $\Gamma \left(\bar{x}\right)\subset U\left(\bar{x}\right)$ in X such that for any sufficiently small γ, the intersection of the set $\Gamma (\bar{x})$ with the ball $B(\bar{x},\gamma )$ is not empty and $\Gamma \left(\bar{x}\right)\cap B\left(\bar{x},\gamma \right)\neq \left\{\bar{x}\right\}$. Moreover, for $x\in \Gamma (\bar{x})$, there exist h(x):X→Y and a constant c such that $0<c_{1}<\infty$ and*(34)$$\Phi_{p}{\left[h\left(x\right)\right]}^{p}=-F\left(x,\bar{y}\right),\;∥h\left(x\right)∥\leq c_{1}\sum_{r=1}^{p}{∥f_{r}\left(x,\bar{y}\right)∥}^{1/r},$$(4)*The Uniform p-Regularity Condition:**The mapping F(x,y) is uniformly p-regular over the set $\Phi_{p}^{-1}\left(-F\left(x,\bar{y}\right)\right).$*
*If conditions 1 to 4 are satisfied, then there exists a constant k>0, a sufficiently small δ≤γ, and a mapping $\varphi :\Gamma \left(\bar{x}\right)\cap B\left(\bar{x},\delta \right)\to U\left(\bar{y}\right)$ such that the following hold for $x\in \Gamma \left(\bar{x}\right)\cap B\left(\bar{x},\delta \right)$:*

$$\varphi \left(\bar{x}\right)=\bar{y};$$


F(x,φ(x))=0,


$$∥\varphi \left(x\right)-\bar{y}∥\leq k\sum_{r=1}^{p}{∥f_{r}\left(x,\bar{y}\right)∥}^{1/r}.$$



There are generalizations of IFT for nonregular mappings derived by other authors. Some examples include a generalization of the IFT and its application to a parametric linear time-optimal control problem presented in [39], generalized IFT applied to ordinary differential equations in [40], and IFT for 2-regular mappings in [41,42].

### 4.3. Newton’s Method

#### 4.3.1. Classical Newton’s Method for Nonlinear Equations and Unconstrained Optimization Problems

Consider the problem of solving the nonlinear Equation (Equation 3), where F:X→Y is sufficiently smooth, so that $F\in C^{p+1}\left(X\right)$ for some p∈N. Let $\bar{x}$ be a solution of (Equation 3), that is, $F(\bar{x})=0$. Assume that mapping *F* is singular at the point $\bar{x}$.

In the finite dimensional case, when $F\left(x\right)={\left(F_{1}\left(x\right),\ldots ,F_{n}\left(x\right)\right)}^{T}$, $X=R^{n}$, and $Y=R^{n}$, the singularity of *F* at $\bar{x}$ means that the Jacobian $F^{\prime }\left(\bar{x}\right)$ of *F* is singular at $\bar{x}$, as in the following example.

**Example** **7** ([43])**.**
*Consider function $F:R^{2}\to R^{2}$ from Example 1, defined by*$$F\left(x\right)=\left(\begin{array}{c} x_{1}+x_{2}\\ x_{1}x_{2} \end{array}\right),$$*where $\bar{x}={\left(0,0\right)}^{T}$ is a solution to Equation *(Equation 3)* and $F^{\prime }\left(\bar{x}\right)=\left(\begin{array}{cc} 1 & 1\\ 0 & 0 \end{array}\right)$ is singular (degenerate) at the point $\bar{x}.$*
*Consider a sufficiently small ε>0 and some initial point $x^{0}\in B\left(0,\varepsilon \right)$. The classical Newton method is defined by*

(35)
$$x^{k+1}=x^{k}-{\left(F^{\prime }\left(x^{k}\right)\right)}^{-1}F\left(x^{k}\right),\;k=0,1,\ldots .$$

*If $x^{k}=\left(x_{1},x_{2}\right)$ in this example, we obtain*

$$F^{\prime }\left(x^{k}\right)=\left(\begin{array}{cc} 1 & 1\\ x_{2} & x_{1} \end{array}\right),\;{\left(F^{\prime }\left(x^{k}\right)\right)}^{-1}=\frac{1}{x_{1}-x_{2}}\left(\begin{array}{cc} x_{1} & -1\\ -x_{2} & 1 \end{array}\right).$$

*Then*

$${\left(F^{\prime }\left(x^{k}\right)\right)}^{-1}F\left(x^{k}\right)=\frac{1}{x_{1}-x_{2}}\left(\begin{array}{c} x_{1}^{2}\\ -x_{2}^{2} \end{array}\right),$$


$$\;x^{k+1}=x^{k}-{\left(F^{\prime }\left(x^{k}\right)\right)}^{-1}F\left(x^{k}\right)=\frac{1}{x_{1}-x_{2}}\left(\begin{array}{c} -x_{1}\;x_{2}\\ x_{1}x_{2} \end{array}\right).$$

*If $x_{1}=x_{2}$, then ${\left(F^{\prime }\left(x^{k}\right)\right)}^{-1}$ does not exist and, hence, method *(Equation 35)* is not applicable. Even in the case when ${\left(F^{\prime }\left(x^{k}\right)\right)}^{-1}$ exists, method *(Equation 35)* might be diverging. As an example, consider point $x^{k}={\left(t+t^{3},t\right)}^{T}$, where t is sufficiently small. Then*$$x^{k+1}=\frac{1}{t^{3}}\left(\begin{array}{c} -t^{2}-t^{4}\\ t^{2}+t^{4} \end{array}\right)={\left(-\frac{1}{t}-t,\frac{1}{t}+t\right)}^{T}$$*and, for a sufficiently small values of t, $∥x^{k+1}-\bar{x}∥=∥x^{k+1}-0∥\approx \frac{1}{t}\to \infty$ when $t\to 0^{+}.$ For instance, if $t=10^{-5}$, then $∥x^{k+1}-0∥\approx 10^{5}$ and the method *(Equation 35)* is diverging.*

For the overview of the existing approaches to Newton-like methods for singular operators, see, e.g., [44].

Now we consider Newton’s method for finding critical points of an unconstrained optimization problem:(36)$$\min_{x\in R^{2}}f\left(x\right),$$
where $f:R^{2}\to R$. The classical Newton’s method applied to problem (Equation 36) has the form(37)$$x^{k+1}=x^{k}-{\left(f^{\prime \prime }\left(x^{k}\right)\right)}^{-1}f^{\prime }\left(x^{k}\right).$$As an example, consider minimization of function *f* given by $f\left(x\right)=x_{1}^{2}+x_{1}^{2}x_{2}+x_{2}^{4}$ (see [43]). One of the critical points of the function *f* is $\bar{x}={\left(0,0\right)}^{T}$. Let $x^{0}={\left(x_{1}^{0},x_{2}^{0}\right)}^{T}$ where $x_{1}^{0}=x_{2}^{0}\sqrt{6(1+x_{2}^{0})}$. Then$$f^{\prime \prime }\left(x^{0}\right)=\left(\begin{array}{cc} 2+2x_{2}^{0} & 2x_{2}^{0}\sqrt{6(1+x_{2}^{0})}\\ 2x_{2}^{0}\sqrt{6(1+x_{2}^{0})} & 12{\left(x_{2}^{0}\right)}^{2} \end{array}\right)$$
and $\det f^{\prime \prime }\left(x^{0}\right)=0.$ Hence, ${\left(f^{\prime \prime }\left(x^{0}\right)\right)}^{-1}$ does not exist, so Newton’s method (Equation 37) is not applicable.

#### 4.3.2. The *p*-Factor Newton’s Method

In this section, we describe a method for solving nonlinear Equation (Equation 3), where $F:R^{n}\to R^{n}$ and the matrix $F^{\prime }\left(\bar{x}\right)$ is singular at the solution point $\bar{x}$ (see [43]). The proposed method is based on the construction of the *p*-factor operator.

There are various publications describing the p-factor-method for solving degenerate nonlinear systems and nonregular optimization problems. Some examples are given in [43,45,46].

Let $h\in R^{n}$. Similarly to the definitions in Section 3, now we define $Y_{1}$ by Y1=ImF′(x¯), and define the projection ${\bar{P}}_{1}=P_{Y_{1}^{⊥}}$ as the projection of *Y* onto the orthogonal complementary subspace $Y_{1}^{⊥}$ of $Y_{1}$ in *Y*. Similarly, we can define $Y_{2}$ asY2=ImF′(x¯)+P¯1F′′(x¯)h,andP¯2=PY2⊥.Continuing in the same way for each k=2,…,p−1, we obtain ${\bar{P}}_{k+1}=P_{Y_{k+1}^{⊥}}$ andYk+1=ImF′(x¯)+∑i=1kP¯iF′′(x¯)h+∑i2>i1i1,i2∈{1,2,3}P¯i2P¯i1F(3)(x¯)[h]2+…$$\left.+\sum_{\begin{array}{c} i_{k}>\ldots >i_{1}\\ i_{1},\ldots ,i_{k}\in \left\{1,\ldots ,k\right\} \end{array}}{\bar{P}}_{i_{k}}\ldots {\bar{P}}_{i_{1}}F^{\left(k\right)}\left(\bar{x}\right){\left[h\right]}^{\left(k-1\right)}\right).$$

Let *h* be a fixed vector such that ∥h∥=1 and mapping *F* is *p*-regular at the solution $\bar{x}$ along vector *h*. Let matrices $P_{i},i=1,\ldots ,p-1$, be defined as follows:$$P_{1}=\sum_{i=1}^{p-1}{\bar{P}}_{i},\;P_{2}=\sum_{\begin{array}{c} i_{2}>i_{1}\\ i_{1},i_{2}\in \left\{1,\ldots ,p-1\right\} \end{array}}{\bar{P}}_{i_{2}}{\bar{P}}_{i_{1}},\;P_{k+1}=\sum_{\begin{array}{c} i_{k}>\ldots >i_{1}\\ i_{1},\ldots ,i_{k}\in \left\{1,\ldots ,p-1\right\} \end{array}}{\bar{P}}_{i_{k}}\ldots {\bar{P}}_{i_{1}}$$
for all k=2,…,p−1.

We assume that $\bar{x}$ is a solution of F(x)=0. Now, instead of $F(\bar{x})=0$, consider$$F\left(\bar{x}\right)+P_{1}F^{\prime }\left(\bar{x}\right)h+\ldots +P_{p-1}F^{\left(p-1\right)}\left(\bar{x}\right){\left[h\right]}^{p-1}=0.$$The assumption of *p*-regularity of the mapping *F* at the solution $\bar{x}$ along the vector *h* implies that the *p*-factor matrix given by(38)$$F^{\prime }\left(\bar{x}\right)+P_{1}F^{\prime \prime }\left(\bar{x}\right)h+\ldots +P_{p-1}F^{\left(p\right)}\left(\bar{x}\right){\left[h\right]}^{p-1}$$
is not singular. Hence, ${\bar{P}}_{p}=0$ and $Y_{p}=R^{n}.$

Then, the *p*-factor Newton method can be defined as(39)$$x^{k+1}=x^{k}-{\left(F^{\prime }\left(x^{k}\right)+P_{1}F^{\prime \prime }\left(x^{k}\right)h+\ldots +P_{p-1}F^{\left(p\right)}\left(x^{k}\right){\left[h\right]}^{p-1}\right)}^{-1}$$$$\;\;\;\times \left(F\left(x^{k}\right)+P_{1}F^{\prime }\left(x^{k}\right)h+\ldots +P_{p-1}F^{\left(p-1\right)}\left(x^{k}\right){\left[h\right]}^{p-1}\right).$$

The following theorem provides conditions that ensure the quadratic convergence of the *p*-factor Newton method (Equation 39).

**Theorem** **12** ([43])**.**
*Let $F\in C^{p}\left(R^{n}\right)$, and let $\bar{x}$ be a solution of F(x)=0. Assume that there exists a vector $h\in R^{n}$, ∥h∥=1, such that the p-factor matrix defined in Equation *(Equation 38)* is not singular. Then, for any $x^{0}\in U_{\varepsilon }\left(\bar{x}\right)$ (with ε>0 sufficiently small) and for the sequence $\left\{x^{k}\right\}$ generated by the method in Equation *(Equation 39)*, the following inequality holds for some constant c>0:*(40)$$∥x^{k+1}-\bar{x}∥\leq c∥x^{k}-\bar{x}{∥}^{2},\;k=0,1,\ldots .$$

In the case of p=2, the *p*-factor Newton method (Equation 39) reduces to the following:(41)$$x^{k+1}=x^{k}-{\left(F^{\prime }\left(x^{k}\right)+P_{1}F^{\prime \prime }\left(x^{k}\right)h\right)}^{-1}\left(F\left(x^{k}\right)+P_{1}F^{\prime }\left(x^{k}\right)h\right)$$
where $P_{1}$ is the orthogonal projection onto Im(F′(x¯))⊥, and the vector *h*(∥h∥=1) is chosen such that the 2-factor matrix(42)$$F^{\prime }\left(\bar{x}\right)+P_{1}F^{\prime \prime }\left(\bar{x}\right)h$$
is not singular. This condition is equivalent to *F* being 2-regular at $\bar{x}$ along *h*. In this case, the equation$$F\left(\bar{x}\right)+P_{1}F^{\prime }\left(\bar{x}\right)h=0$$
is satisfied at $\bar{x}$. Note that (Equation 42) implies that $\bar{x}$ is a locally unique solution of (Equation 3).

The 2-factor Newton method presented here can be applied to solve the equation in Example 7. Specifically, instead of using the iterative procedure (Equation 35), the 2-factor Newton method given by (Equation 41) should be used.

**Example** **8** ([43])**.**
*Consider the following problem*$$\min_{x\in R^{2}}\;f\left(x\right),$$*where $f:R^{2}\to R$ is defined by $f\left(x\right)=x_{1}^{2}+x_{1}^{2}x_{2}+x_{2}^{4}$. If $F\left(x\right)=f^{\prime }\left(x\right)$, then $F\left(x\right)=\left(\begin{array}{c} 2x_{1}+2x_{1}\;x_{2}\\ x_{1}^{2}+4x_{2}^{3} \end{array}\right)$, and for $\bar{x}={\left(0,0\right)}^{T}$, we have $F\left(0,0\right)={\left(0,0\right)}^{T}.$ It can be shown that F is 3-regular at (0,0) along $h={\left(1,1\right)}^{T}$.*
*Namely, according to the previous definitions, in this example,*

$${\bar{P}}_{1}=\left(\begin{array}{cc} 0 & 0\\ 0 & 1 \end{array}\right),\;{\bar{P}}_{2}=\frac{1}{2}\left(\begin{array}{cc} 1 & -1\\ -1 & 1 \end{array}\right),$$


P1=P¯1+P¯2=121−1−13,P2=P¯2P¯1=1200−11.

*Then, the following matrix is nonsingular:*

F′(0)+P1F′′(0)h+P2F(3)(0)[h]2=f′′(0)+P1f(3)(0)h+P2f(4)(0)[h]2=22−1111.


*Consider the 3-factor method:*

$$\begin{array}{c} x^{k+1}=x^{k}-\\ {\left(f^{\prime \prime }\left(0\right)+P_{1}f^{\left(3\right)}\left(0\right)\left[h\right]+P_{2}f^{\left(4\right)}\left(0\right){\left[h\right]}^{2}\right)}^{-1}\left(f^{\prime }\left(x^{k}\right)+P_{1}f^{\prime \prime }\left(x^{k}\right)\left[h\right]+P_{2}f^{\left(3\right)}\left(x^{k}\right){\left[h\right]}^{2}\right). \end{array}$$


*Let $x^{k}={\left(x_{1},x_{2}\right)}^{T}.$ Then*

$$\begin{array}{cc} ∥x^{k+1}-0∥ & =∥x^{k}-{\left(\begin{array}{cc} 2 & -11\\ 2 & 11 \end{array}\right)}^{-1}\left(\begin{array}{c} 2x_{1}-11x_{2}+2x_{1}x_{2}-6x_{2}^{2}\\ 2x_{1}+11x_{2}+x_{1}^{2}+18x_{2}^{2}+4x_{2}^{3} \end{array}\right)∥=\\ & =\frac{1}{44}∥\left(\begin{array}{c} 11x_{1}^{2}+132x_{2}^{2}+22x_{1}x_{2}+44x_{2}^{3}\\ 2x_{1}^{2}+48x_{2}^{2}-4x_{1}x_{2}+8x_{2}^{3} \end{array}\right)∥\leq 10{∥x^{k}-0∥}^{2}. \end{array}$$



### 4.4. Optimality Conditions for Equality-Constrained Optimization Problems

In this section, we consider optimization problem (Equation 4):minf(x)subjecttoF(x)=0,
where f:X→R is a sufficiently smooth function and F:X→Y is a sufficiently smooth mapping from a Banach space *X* to a Banach space *Y*.

#### 4.4.1. Optimality Conditions:
Lagrange Multiplier Theorem

There is an extensive body of literature discussing optimality conditions for regular constrained optimization problems, which are problems that satisfy certain constraint qualifications. One notable reference on this topic is Chapter 3 of the book [47].

The classical optimality conditions state that if $\bar{x}$ is a regular solution of Problem (Equation 4), then there exists a Lagrange multiplier in the form of a constant vector $\bar{\lambda }\in Y^{*}$, such that(43)$$f^{\prime }\left(\bar{x}\right)=F^{\prime }{\left(\bar{x}\right)}^{*}\bar{\lambda },$$
where $F^{\prime }{\left(\bar{x}\right)}^{*}:Y^{*}\to X^{*}$ denotes the adjoint of $F^{\prime }\left(\bar{x}\right)$, and $X^{*}$ and $Y^{*}$ denote the dual spaces of *X* and *Y*, respectively.

The situation changes in the degenerate case when the derivative $F\prime \left(\bar{x}\right)$ is not surjective. In such cases, the classical optimality conditions in the form of Equation (Equation 43) do not hold, as illustrated in the following example.

**Example** **9.** *Consider the problem*(44)minimizex∈R3x22+x3subjecttox12−x22+x32x12−x22+x32+x2x3=(00).*Note that mapping $F\left(x\right)=\left(\begin{array}{c} x_{1}^{2}-x_{2}^{2}+x_{3}^{2}\\ x_{1}^{2}-x_{2}^{2}+x_{3}^{2}+x_{2}x_{3} \end{array}\right)$ was introduced in* (Equation 28)*.**In this example, if $\bar{x}={\left(0,0,0\right)}^{T}$, then*f′(x¯)=(0,0,1)T,andF′(x¯)=000000.*Hence $f^{\prime }\left(\bar{x}\right)\neq F^{\prime }{\left(\bar{x}\right)}^{T}\bar{\lambda }$ and Equation* (Equation 43) *does not hold.*

#### 4.4.2. Optimality Conditions for *p*-Regular Optimization Problems

In this section, we will focus on the case when the equality constraints defined by mapping F(x) are not regular at a solution $\bar{x}$ of the problem (Equation 4). We define the *p*-factor-Lagrange function $L_{p}\left(x,\lambda \left(h\right),h\right):X\times \left(Y_{1}^{*}\times \ldots \times Y_{p}^{*}\right)\times X\to R$ as(45)$$L_{p}\left(x,\lambda \left(h\right),h\right)=f\left(x\right)+\sum_{i=1}^{p}\langle \lambda_{i}\left(h\right),\;f_{i}^{\left(i-1\right)}\left(x\right){\left[h\right]}^{i-1}\rangle ,$$
where x,h∈X, $\lambda_{i}\left(h\right)\in Y_{i}^{*}$ for i=1,…,p, and the mappings $f_{i}\left(x\right)$ are defined in (Equation 16). Note that the *p*-factor-Lagrange function is a generalization of the classical Lagrange function and reduces to it in the regular case.

The development of optimality conditions for nonregular problems has become an active area of research (see [16,48,49,50,51] and references therein).

To state the sufficient conditions in Theorem 13, we also introduce an alternative version of the *p*-factor-Lagrange function, ${\bar{L}}_{p}(x,\lambda \left(h\right),h:X\times \left(Y_{1}^{*}\times \ldots \times Y_{p}^{*}\right)\times X\to R,$ which is defined as follows:(46)$${\bar{L}}_{p}\left(x,\lambda \left(h\right),h\right)=f\left(x\right)+\sum_{i=1}^{p}\frac{2}{i(i+1)}\langle \lambda_{i}\left(h\right),\;f_{i}^{\left(i-1\right)}\left(x\right){\left[h\right]}^{i-1}\rangle .$$

To state optimality conditions for *p*-regular optimization problems, we use the definition of strong *p*-regularity at $\bar{x}$ given in Definition 7. We also use the set $H_{p}\left(\bar{x}\right)$ defined in Equation (Equation 23), and the operator $\Psi_{p}\left(h\right)$ defined in Equation (Equation 19).

**Theorem** **13** ([16], necessary and sufficient conditions for optimality)**.**
*Assume that X and Y are Banach spaces, U is a neighborhood of a point $\bar{x}$ in X, f:U→R is a twice continuously Fréchet differentiable function in U, and F:U→Y is a (p+1)-times Fréchet differentiable mapping in U.*
*Necessary conditions for optimality.*
*Assume that for an element $h\in H_{p}\left(\bar{x}\right)$, the set $Im\;\Psi_{p}\left(h\right)$ is closed in $Y_{1}⊕\ldots ⊕Y_{p}$. Suppose that F is p-regular at the point $\bar{x}$ along the vector $h\in H_{p}\left(\bar{x}\right)$. If $\bar{x}$ is a local minimizer of problem* (Equation 4)*, then there exist multipliers $\bar{\lambda }\left(h\right)=\left({\bar{\lambda }}_{1}\left(h\right),\ldots ,{\bar{\lambda }}_{p}\left(h\right)\right)\in \left(Y_{1}^{*}\times \ldots \times Y_{p}^{*}\right)$ such that the partial derivative of the function ${\bar{L}}_{p}$ with respect to x, denoted by ${\left(L_{p}^{\prime }\right)}_{x}$, satisfies*
(47)$${\left(L_{p}^{\prime }\right)}_{x}\left(\bar{x},\bar{\lambda }\left(h\right),h\right)=0.$$
*Sufficient conditions for optimality.*
*Assume that the set ImΨp(h) is closed in $Y_{1}⊕\ldots ⊕Y_{p}$ for every $h\in H_{p}\left(\bar{x}\right)$, and that ImΨp(h)=Y1⊕…⊕Yp. Assume also that the mapping F is strongly p-regular at $\bar{x}.$ Suppose that there exist a constant α>0 and a multiplier $\bar{\lambda }\left(h\right)$ such that Equation* (Equation 47) *is satisfied, and that the second-order partial derivative of the function ${\bar{L}}_{p}$ (defined in* (Equation 46)) *with respect to x, denoted by ${\left({\bar{L}}_{p}\right)}_{xx}$, satisfies*
(48)$${\left({\bar{L}}_{p}\right)}_{xx}\left(\bar{x},\bar{\lambda }\left(h\right),h\right){\left[h\right]}^{2}\geq \alpha {∥h∥}^{2}$$
*for every $h\in H_{p}\left(\bar{x}\right).$ Then $\bar{x}$ is a strict local minimizer of the problem Equation* (Equation 4)*.*

 **Example** **10.** *In this example, we continue with the analysis of problem* (Equation 44) *from Section 4.4.1. It can be verified that the point $\bar{x}=\left({\bar{x}}_{1},{\bar{x}}_{2},{\bar{x}}_{3}\right)={\left(0,0,0\right)}^{T}$ is a local minimizer of Equation* (Equation 44)*. In Example 6, we showed that the mapping $F\left(x\right)=\left(\begin{array}{c} x_{1}^{2}-x_{2}^{2}+x_{3}^{2}\\ x_{1}^{2}-x_{2}^{2}+x_{3}^{2}+x_{2}x_{3} \end{array}\right)$ is 2-regular at $\bar{x}$ along the vector $h={\left(1,1,0\right)}^{T}$.**In this example, the 2-factor Lagrange function $L_{2}$, defined in* (Equation 45) *for p=2, is given by*
$$L_{2}\left(x,\lambda \left(h\right),h\right)=f\left(x\right)+\langle \lambda_{1}\left(h\right),\;f_{1}\left(x\right)\rangle +\langle \lambda_{2}\left(h\right),\;f_{2}^{\prime }\left(x\right)\left[h\right]\rangle ,$$
*where $\lambda \left(h\right)=(\lambda_{1}\left(h\right),\lambda_{2}\left(h\right))$, $\lambda_{1}\left(h\right)={\left(0,0\right)}^{T}$, and $\lambda_{2}\left(h\right)={\left(\alpha ,\beta \right)}^{T}.$ Substituting the given expressions, we obtain the following form:*
$$L_{2}\left(x,\lambda \left(h\right),h\right)=x_{2}^{2}+x_{3}+\alpha \left(x_{1}-x_{2}\right)+\beta \left(x_{1}-x_{2}+\frac{1}{2}x_{3}\right).$$*Solving the equation*$$L_{2\;x}^{\prime }\left(\bar{x},\lambda \left(h\right),h\right)=0,$$*we obtain the following system:*$$\begin{array}{ccc} \alpha +\beta & = & 0\\ 2{\bar{x}}_{2}-\alpha -\beta & = & 0\\ 1+\frac{1}{2}\beta & = & 0. \end{array}$$*Substituting ${\bar{x}}_{2}=0$, we obtain β=−2 and α=2. Hence, the function ${\bar{L}}_{2}\left(x,\lambda \left(h\right),h\right),$ defined in* (Equation 46)*, takes the following form in this example:*
$${\bar{L}}_{2}\left(x,\lambda \left(h\right),h\right)=x_{2}^{2}+x_{3}+\frac{2}{3}\left(x_{1}-x_{2}\right)-\frac{2}{3}\left(x_{1}-x_{2}+\frac{1}{2}x_{3}\right)=x_{2}^{2}+\frac{2}{3}x_{3}.$$
*Recall that the set $H_{2}\left(\bar{x}\right)$ was determined in Example 6 as*

$$H_{2}\left(\bar{x}\right)={\mathrm{Ker}}^{2}F^{\prime \prime }\left(\bar{x}\right)=\mathrm{span}\left(\left[\begin{array}{c} 1\\ -1\\ 0 \end{array}\right]\right)⋃\mathrm{span}\left(\left[\begin{array}{c} 1\\ 1\\ 0 \end{array}\right]\right).$$

*Then, the second derivative of $\bar{L}$, defined in* (Equation 46) *for p=2, satisfies*
$${\left({\bar{L}}_{2}^{\prime \prime }\right)}_{xx}\left(\bar{x},\lambda \left(h\right),h\right){\left[h\right]}^{2}=2\geq \alpha {∥h∥}^{2}$$
*for some α>0, and for every $h\in H_{2}\left(\bar{x}\right)$. Hence, the sufficient conditions in Theorem 13 are satisfied, and we conclude that $\bar{x}$ is a strict local minimizer of problem Equation* (Equation 44)*.*

### 4.5. Modified Lagrangian Function Method

#### 4.5.1. The Problem

Consider the following constrained optimization problem:(49)$$\min f\left(x\right),\;\mathrm{subject}\;\mathrm{to}\;\;g_{i}\left(x\right)\leq 0,\;i=1,\ldots ,m,$$
where $f:R^{n}\to R$ is an objective function, and $g_{i}:R^{n}\to R$ are constraint functions. The goal is to find a vector $(\bar{x}\in R^{n}$ such that f(x) is minimized while satisfying all constraints. To solve this, we introduce the modified Lagrangian function $L_{E}\left(x,\lambda \right):R^{n}\times R^{m}\to R$, which incorporates both the objective function and the constraints (see, e.g., [45,52,53]):(50)$$L_{E}\left(x,\lambda \right)=f\left(x\right)+\frac{1}{2}\sum_{i=1}^{m}\lambda_{i}^{2}g_{i}\left(x\right),$$
where $\lambda =(\lambda_{1},\ldots ,\lambda_{m})$. This modified Lagrangian function transforms the nonlinear optimization problem into a system of nonlinear equations.

Define the mapping $G:R^{n}\times R^{m}\to R^{n+m}$ by(51)$$G\left(x,\lambda \right)=\left(\begin{array}{c} \nabla f\left(x\right)+\frac{1}{2}\sum_{i=1}^{m}\lambda_{i}^{2}\nabla g_{i}\left(x\right)\\ D(\lambda )g(x) \end{array}\right),$$
where $D\left(\lambda \right)=\mathrm{diag}\left\{\lambda_{i}\right\},$i=1,…,m, and $\lambda \in R^{m}$.

Consider the equation(52)G(x,λ)=0.Let $g^{\prime }\left(x\right)$ be the Jacobian matrix of the mapping g(x). Then, the Jacobian matrix $G^{\prime }\left(x,\lambda \right)$ of the mapping G(x,λ) is given by(53)$$G^{\prime }\left(x,\lambda \right)=\left(\begin{array}{cc} \nabla^{2}f\left(x\right)+\frac{1}{2}\sum_{i=1}^{m}\lambda_{i}^{2}g_{i}^{\prime \prime }\left(x\right) & {\left(g^{\prime }\left(x\right)\right)}^{T}D\left(\lambda \right)\\ D\left(\lambda \right){\left(g^{\prime }\left(x\right)\right)}^{T} & D(g(x)) \end{array}\right).$$

Define the set $I\left(\bar{x}\right)=\left\{j=1,\ldots ,m|g_{j}\left(\bar{x}\right)=0\right\}$ consisting of active constraints, and the set$$I_{0}\left(\bar{x}\right)=\left\{j=1,\ldots ,m|{\bar{\lambda }}_{j}=0,g_{j}\left(\bar{x}\right)=0\right\}\subset I\left(\bar{x}\right),$$
consisting of weakly active constraints, and the set $I_{+}\left(\bar{x}\right)=I\left(\bar{x}\right)\setminus I_{0}\left(\bar{x}\right)$, consisting of strongly active constraints.

Recall that the Strict Complementary Condition (SCQ) means that, for each index j=1,…,m, one and only one of $g_{j}\left(\bar{x}\right)$ and ${\bar{\lambda }}_{j}$ is equal to zero. If $\left(\bar{x},\bar{\lambda }\right)$ is a solution of Problem (Equation 52), and for some index *j*, both $g_{i}\left(\bar{x}\right)=0$ and ${\bar{\lambda }}_{i}=0$, then the set $I_{0}\left(\bar{x}\right)$ is nonempty, and the SCQ fails. Consequently, $G^{\prime }\left(\bar{x},\bar{\lambda }\right)$ is a degenerate matrix. Example 11 illustrates this situation.

**Example** **11** ([45])**.**
*Consider the problem*(54)$$\min_{x\in R^{n}}\left(x_{1}^{2}+x_{2}^{2}+4x_{1}x_{2}\right)\;\mathrm{subject}\;\mathrm{to}\;\;x_{1}\geq 0,\;x_{2}\geq 0.$$
*A direct argument confirms that $\bar{x}={\left(0,0\right)}^{T}$ is a solution of Problem* (Equation 54) *with the corresponding Lagrange multiplier $\bar{\lambda }={\left(0,0\right)}^{T}$.**The modified Lagrange function in this case is*$$L_{E}\left(x,\lambda \right)=x_{1}^{2}+x_{2}^{2}+4x_{1}x_{2}-\frac{1}{2}\lambda_{1}^{2}x_{1}-\frac{1}{2}\lambda_{2}^{2}x_{2}.$$*The mapping G is defined by*$$G\left(x,\lambda \right)=\left(\begin{array}{c} 2x_{1}+4x_{2}-\frac{1}{2}\lambda_{1}^{2}\\ 2x_{2}+4x_{1}-\frac{1}{2}\lambda_{2}^{2}\\ -\lambda_{1}x_{1}\\ -\lambda_{2}x_{2} \end{array}\right),$$*and, therefore, the Jacobian matrix $G^{\prime }\left(\bar{x},\bar{\lambda }\right)$ defined in* (Equation 53) *is singular.*

#### 4.5.2. Modified Lagrange Function Method for 2-Regular Problems

In this section, we consider the constrained optimization problem (Equation 49) with the modified Lagrangian function $L_{E}\left(x,\lambda \right)$ defined in (Equation 50). We focus on the nonregular case when the Jacobian matrix $G^{\prime }\left(\bar{x},\bar{\lambda }\right)$ defined in (Equation 53) is singular at the solution $\left(\bar{x},\bar{\lambda }\right)$ of (Equation 52).

We will show that the mapping G(x,λ) defined in (Equation 51) is 2-regular at $\left(\bar{x},\bar{\lambda }\right)$.

Without loss of generality, assume that $I_{0}\left(\bar{x}\right)=\left\{1,\ldots ,s\right\}$, so that ${\bar{\lambda }}_{j}=0$ and $g_{j}\left(\bar{x}\right)=0$ for all j=1,…,s. Additionally, we assume that $I_{+}\left(\bar{x}\right)=\left\{s+1,s+2,\ldots ,p\right\}$. Introduce the notation l=m−p. Then, the rows of matrix $G^{\prime }\left(\bar{x},\bar{\lambda }\right)$ with the numbers from the (n+1)th to the (n+s)th contain only zeros. Define the vector $h\in R^{n+m}$ as follows(55)$$h^{T}=\left(0_{n}^{T},1_{s}^{T},0_{m-s}^{T}\right),$$
where $1_{s}^{T}$ is an *s*-dimensional all-one row vector.

Let the mapping $\Phi :R^{n}\times R^{m}$ be given by(56)$$\Phi \left(x,\lambda \right)=G\left(x,\lambda \right)+G^{\prime }\left(x,\lambda \right)h,$$
where *h* is defined in (Equation 55).

The following result is well known.

**Lemma** **3** ([45])**.**
*Let an n×n matrix V and an n×p matrix Q satisfy the properties:*
 *1.* *Q has linearly independent columns, and* *2.* *$x^{T}Vx>0$ for all $x\in \mathrm{Ker}Q^{T}\setminus \left\{0\right\}$.*
*Assume also that $D_{N}$ is a full-rank diagonal l×l matrix. Then, the matrix $\bar{A}$ defined by*

(57)
$$\bar{A}=\left(\begin{array}{ccc} V & Q & 0\\ Q^{T} & 0 & 0\\ 0 & 0 & D_{N} \end{array}\right)$$

*is a nonsingular matrix.*


The Linear Independence Constraint Qualification (LICQ) holds for the optimization problem (Equation 49) if the gradients of active constraints are linearly independent.

The second-order sufficient optimality condition holds if there exists α>0 such that(58)$$z^{T}\nabla_{xx}^{2}L_{E}\left(\bar{x},\bar{\lambda }\right)z\geq \alpha {∥z∥}^{2}$$
for all $z\in R^{n}$ that satisfy the conditions$${\left(\nabla g_{j}\left(\bar{x}\right)\right)}^{T}z\leq 0\;\forall j\in I\left(\bar{x}\right).$$

**Lemma** **4** ([45])**.**
*Let $f,g_{i}\in C^{3}\left(R^{n}\right)$, for i=1,…,m. Assume that the LICQ (Linear Independence Constraint Qualification) and the second-order sufficient optimality conditions are satisfied at the solution $\left(\bar{x},\bar{\lambda }\right)$ of *(Equation 52)*, and that Φ is a mapping given by Equation *(Equation 56)*. Then, the 2-factor operator*$$\Phi^{\prime }\left(x,\lambda \right)=G^{\prime }\left(x,\lambda \right)+G^{\prime \prime }\left(x,\lambda \right)h$$*is nonsingular at the point $\left(\bar{x},\bar{\lambda }\right)$.*

The proof of Lemma 4 can be derived from Lemma 3.

Indeed, if D(λ) is a diagonal matrix with $\lambda_{j}$ as the *j*-th diagonal entry,$$V=\nabla_{xx}^{2}L_{E}\left(\bar{x},\bar{\lambda }\right),\;D_{N}=D\left(g_{N}\left(\bar{x}\right)\right),\;g_{N}\left(x\right)={\left(g_{p+1}\left(x\right),\ldots ,g_{m}\left(x\right)\right)}^{T},$$
and$$Q=\left[\nabla g_{1}\left(\bar{x}\right),\ldots ,\nabla g_{s}\left(\bar{x}\right),{\bar{\lambda }}_{s+1}\nabla g_{s+1}\left(\bar{x}\right),\ldots ,{\bar{\lambda }}_{p}\nabla g_{p}\left(\bar{x}\right)\right],$$
then $\Phi^{\prime }\left(\bar{x},\bar{\lambda }\right)=\bar{A}$, where matrix $\bar{A}$ is defined in (Equation 57). Lemma 4 implies that the 2-factor Newton method is given by(59)$$w^{k+1}=w^{k}-{\left(G^{\prime }\left(w^{k}\right)+G^{\prime \prime }\left(w^{k}\right)h\right)}^{-1}\left(G\left(w^{k}\right)+G^{\prime }\left(w^{k}\right)h\right),\;k=0,1,\ldots ,$$
and it can be applied to solve the system (Equation 52), where *G* is defined in (Equation 51). As a result, we have the following theorem.

**Theorem** **14** ([45])**.**
*Let $\bar{x}$ be a solution to *(Equation 49)* and $f,g_{i}\left(x\right)\in C^{3}\left(R^{n}\right)$, for i=1,…,m. Assume that the LICQ and the second-order sufficient optimality conditions *(Equation 58)* are satisfied at the point $\bar{x}$. Then, there exists a sufficiently small open ball $B(\bar{w},\varepsilon )$, where $\bar{w}=\left(\bar{x},\bar{\lambda }\right)$, such that the estimate*$$∥w^{k+1}-\bar{w}∥\leq \beta ∥w^{k}-\bar{w}{∥}^{2},$$*holds for the method *(Equation 59)*, where $w^{0}\in B\left(\bar{w},\varepsilon \right)$ and β>0 is a constant independent of k.*

In addition, there are other publications where a modified Lagrange function is used in various contexts, such as [54,55]. Higher-order analysis of optimality conditions has been performed in [56].

### 4.6. Calculus of Variations

The methods of the calculus of variations are widely used to solve many problems in physics and classical mechanics. However, since the classical approach cannot be directly applied to many of these problems, there is a need to extend or reformulate classical theorems to accommodate irregular cases. Over the years, various types of irregular problems in the calculus of variations have been extensively studied in both mathematics and its applications (see, e.g., [1,3,32,57,58,59,60,61,62]).

#### 4.6.1. Singular Problems of Calculus of Variations

In this section, we consider the following Lagrange problem, which involves finding a curve x=x(t), such that (see [63]):(60)$$J_{0}\left(x\right)=\int_{t_{1}}^{t_{2}}f\left(t,x,\dot{x}\right)dt\to \min$$
subject to the subsidiary conditions:(61)$$\Gamma \left(x\right)=0,\;q(x\left(t_{1}\right),x\left(t_{2}\right))=0,$$
where$$\Gamma \left(x\right)\left(t\right)=G\left(t,x\left(t\right),\dot{x}\left(t\right)\right)=0\;forallt\in \left[t_{1},t_{2}\right],\;\dot{x}=\frac{dx}{dt},$$$$X=C_{n}^{1}\left(\left[t_{1},t_{2}\right]\right),\;Y=C_{m}\left(\left[t_{1},t_{2}\right]\right),\;x\left(t\right)\in X,\;\Gamma \in C^{p+1}\left(X,Y\right),$$$$G:R\times R^{n}\times R^{n}\to R^{m},\;G\left(t,x\left(t\right),\dot{x}\left(t\right)\right)=(G_{1}\left(t,x\left(t\right),\dot{x}\left(t\right)\right),\ldots ,G_{m}\left(t,x\left(t\right),\dot{x}\left(t\right)\right),$$
and$$q:R^{n}\times R^{n}\to R^{k},\;f:R\times R^{n}\times R^{n}\to R.$$We assume that all mappings and their partial derivatives are continuous with respect to t,x, and $\dot{x}$. We denote by $\bar{x}\left(t\right)$ a solution to Problem (Equation 60) and (Equation 61). While each of *x*, $\dot{x}$, and each component of *x* is a function of *t*, (e.g., x=x(t), and $\dot{x}=\dot{x}(t)$), we do not write this explicitly in order to avoid over complicated notation.

In the regular case, when ImΓ′(x¯)=Y, the Euler–Lagrange necessary conditions are satisfied and take the form (see, e.g., [60,64]):(62)$$f_{x}+\lambda \left(t\right)G_{x}-\frac{d}{dt}\left(f_{\dot{x}}+\lambda \left(t\right)G_{\dot{x}}\right)=0\;\mathrm{for}\;\mathrm{all}\;t\in \left[t_{1},t_{2}\right].$$Let $\lambda \left(t\right)={\left(\lambda_{1}\left(t\right),\ldots ,\lambda_{m}\left(t\right)\right)}^{T}.$ Then$$\lambda \left(t\right)G=\lambda_{1}\left(t\right)G_{1}+\ldots +\lambda_{m}\left(t\right)G_{m}\;and\;\lambda \left(t\right)G_{x}=\lambda_{1}\left(t\right)G_{1x}+\ldots +\lambda_{m}\left(t\right)G_{mx}.$$

In the singular case, when ImΓ′(x¯)≠Y, we can only guarantee that the following equation is satisfied:(63)$$\lambda_{0}f_{x}+\lambda \left(t\right)G_{x}-\frac{d}{dt}\left(\lambda_{0}f_{\dot{x}}+\lambda \left(t\right)G_{\dot{x}}\right)=0,$$
where $\lambda_{0}^{2}+{∥\lambda \left(t\right)∥}^{2}=1.$ In this case, $\lambda_{0}$ might be equal to 0, which results in no constructive conditions for the description or finding $\bar{x}\left(t\right)$.

**Example** **12** ([63])**.**
*Consider the following problem of finding a curve $x\left(t\right)=(x_{1}\left(t\right),\ldots ,x_{5}\left(t\right))$ such that*(64)$$J_{0}\left(x\right)=\int_{0}^{2\pi }\left(x_{1}^{2}+x_{2}^{2}+x_{3}^{2}+x_{4}^{2}+x_{5}^{2}\right)dt\to \min$$*subject to*
(65)$$\Gamma \left(x\right)=\left(\begin{array}{c} \dot{x_{1}}-x_{2}+x_{3}^{2}x_{1}+x_{4}^{2}x_{2}-x_{5}^{2}\left(x_{1}+x_{2}\right)\\ \dot{x_{2}}+x_{1}+x_{3}^{2}x_{2}-x_{4}^{2}x_{1}-x_{5}^{2}\left(x_{2}-x_{1}\right) \end{array}\right)=0,$$$$x_{i}\left(0\right)=x_{i}\left(2\pi \right),\;i=1,\ldots ,5,$$*where $\Gamma :C_{5}^{2}\left(\left[0,2\pi \right]\right)\to C_{2}\left(\left[0,2\pi \right]\right),$ ${\dot{x}}_{1}=\frac{dx_{1}}{dt}$, and ${\dot{x}}_{2}=\frac{dx_{2}}{dt}.$**Here,*$$f\left(x\right)=x_{1}^{2}+x_{2}^{2}+x_{3}^{2}+x_{4}^{2}+x_{5}^{2},\;and\;q_{i}\left(x\left(0\right),x\left(2\pi \right)\right)=x_{i}\left(0\right)-x_{i}\left(2\pi \right),\;i=1,\ldots ,5.$$*The solution of Problem *(Equation 64)* and *(Equation 65)* is $\bar{x}\left(t\right)=0$ and $\Gamma^{\prime }\left(0\right)$ is singular. Indeed, using the differentiation rules in functional spaces, we obtain*$$\Gamma^{\prime }\left(0\right)=\left(\begin{array}{ccccc} {\dot{\left(\cdot \right)}}_{1} & -{\left(\cdot \right)}_{2} & 0 & 0 & 0\\ {\left(\cdot \right)}_{1} & {\dot{\left(\cdot \right)}}_{2} & 0 & 0 & 0 \end{array}\right)\;and\;\Gamma^{\prime }\left(0\right)x=\left(\begin{array}{c} {\dot{x}}_{1}-x_{2}\\ x_{1}+{\dot{x}}_{2} \end{array}\right).$$
*Introducing the notation $z\left(t\right)=x_{1}\left(t\right)$ and using the methods of differential equations, one can show that the mapping $\Gamma \left(z\left(t\right)\right)=z^{\prime \prime }\left(t\right)+z\left(t\right),$ with boundary conditions z(0)=z(2π), is not surjective. Indeed, for y∈C[0,2π], satisfying*

$$\int_{0}^{2\pi }\sin\tau \;y\left(\tau \right)d\tau \neq 0\;\;or\;\;\int_{0}^{2\pi }\cos\tau \;y\left(\tau \right)d\tau \neq 0,$$

*the equation $z^{\prime \prime }\left(t\right)+z\left(t\right)=y\left(t\right)$ does not have a solution.*
*With z=z(t), the corresponding Euler–Lagrange equations in this case are as follows: *$$\begin{array}{ccc} 2\lambda_{0}z+\lambda_{2}-{\dot{\lambda }}_{1}+\lambda_{1}x_{3}^{2}+\lambda_{2}x_{5}^{2}-\lambda_{2}x_{4}^{2} & = & 0,\\ 2\lambda_{0}x_{2}-\lambda_{1}-{\dot{\lambda }}_{2}+\lambda_{1}x_{4}^{2}+\lambda_{2}x_{3}^{2}-\lambda_{1}x_{5}^{2}-\lambda_{2}x_{5}^{2} & = & 0,\\ 2\lambda_{0}x_{3}+2\lambda_{1}zx_{3}+2\lambda_{2}x_{2}x_{3} & = & 0,\\ 2\lambda_{0}x_{4}+2\lambda_{1}x_{2}x_{4}-\lambda_{2}zx_{4} & = & 0,\\ 2\lambda_{0}x_{5}-2\lambda_{1}x_{5}z-2\lambda_{1}x_{2}x_{5}-2\lambda_{2}x_{2}x_{5}+2\lambda_{2}zx_{5} & = & 0,\\ \lambda_{i}\left(0\right)=\lambda_{i}\left(2\pi \right),\;\;i=1,2. & \end{array}$$*Unfortunately, we cannot guarantee that $\lambda_{0}\neq 0$. For $\lambda_{0}=0$, we obtain a series of spurious solutions to the problem *(Equation 64)* and *(Equation 65)*:*(66)$$z\left(t\right)=a\sin t,\;x_{2}=a\cos t,\;x_{3}=x_{4}=x_{5}=0,\;\lambda_{1}=b\sin t,\;\lambda_{2}=b\cos t,\;a,b\in R.$$*The derivation of the solutions *(Equation 66)* is based on standard techniques, so we are omitting the technical details from the paper.*

#### 4.6.2. Optimality Conditions for *p*-Regular Problems of Calculus of Variations

To formulate optimality conditions for the problem (Equation 60) and (Equation 61) in the singular case, we define the *p*-factor Euler–Lagrange function by$$E\left(x\right)=f\left(x\right)+\lambda \left(t\right)\Gamma^{\left(p-1\right)}\left(x\right){\left[h\right]}^{p-1},$$
where$$\Gamma^{\left(p-1\right)}\left(x\right){\left[h\right]}^{p-1}=g_{1}\left(x\right)+g_{2}^{\prime }\left(x\right)\left[h\right]+\ldots +g_{p}^{\left(p-1\right)}\left(x\right){\left[h\right]}^{p-1},$$$$\lambda \left(t\right)\Gamma^{\left(p-1\right)}\left(x\right){\left[h\right]}^{p-1}=\langle \lambda \left(t\right),\left(g_{1}\left(x\right)+g_{2}^{\prime }\left(x\right)\left[h\right]+\ldots +g_{p}^{\left(p-1\right)}\left(x\right){\left[h\right]}^{p-1}\right)\rangle ,$$$$\lambda \left(t\right)={\left(\lambda_{1}\left(t\right),\ldots ,\lambda_{m}\left(t\right)\right)}^{T},\;h=h\left(t\right)\in X.$$

Functions $g_{i}\left(x\right),$i=1,…,p, are determined for the mapping Γ(x) in a way that is similar to how functions $f_{i}\left(x\right),$i=1,…,p, are defined for the mapping F(x), in Equation (Equation 16). Namely,$$g_{k}\left(x\right)=P_{Y_{k}}\Gamma \left(x\right),\;k=1,\ldots ,p.$$

Let$$g_{k}^{\left(k-1\right)}\left(x\right){\left[h\right]}^{k-1}=\sum_{i+j=k-1}C_{k-1}^{i}{g_{k}^{\left(k-1\right)}}_{x^{i}{\left(\dot{x}\right)}^{j}}\left(x\right){\left[h\right]}^{i}{\left[h^{\prime }\right]}^{j},\;k=1,\ldots ,p,$$
where ${g_{k}^{\left(k-1\right)}}_{x^{i}{\left(\dot{x}\right)}^{j}}\left(x\right)={g_{k}^{\left(k-1\right)}}_{\underset{i}{\underset{︸}{x\ldots x}}\underset{j}{\underset{︸}{\dot{x}\ldots \dot{x}}}}\left(x\right).$

**Definition** **10.** *Let $X=C_{n}^{1}\left(\left[t_{1},t_{2}\right]\right)$ and $Y=C_{m}\left(\left[t_{1},t_{2}\right]\right)$. We say that problem *(Equation 60)* and * (Equation 61)* is p-regular at $\bar{x}\left(t\right)\in X$ along some vector h(t)∈X, $h\left(t\right)\in \overset{p}{\underset{k=1}{⋂}}{\mathrm{Ker}}^{k}g_{k}^{\left(k\right)}\left(\bar{x}\left(t\right)\right),$∥h(t)∥≠0, if*
Img1′(x¯(t))+…+gp(p)(x¯(t))[h(t)]p−1=Y.

**Theorem** **15** ([63])**.**
*Assume that the problem *(Equation 60)* and * (Equation 61)* is p-regular at its solution $\bar{x}\left(t\right)\in X$ along h=h(t)∈X, $h\in \overset{p}{\underset{k=1}{⋂}}{\mathrm{Ker}}^{k}g_{k}^{\left(k\right)}\left(\bar{x}\left(t\right)\right).$ Then, there exists a multiplier $\hat{\lambda }\left(t\right)={\left({\hat{\lambda }}_{1}\left(t\right),\ldots ,{\hat{\lambda }}_{m}\left(t\right)\right)}^{T}$ such that the following p-factor Euler–Lagrange equation holds:*(67)$$\begin{array}{c} E_{x}\left(\bar{x}\left(t\right)\right)-\frac{d}{dt}E_{\dot{x}}\left(\bar{x}\left(t\right)\right)=f_{x}\left(\bar{x}\left(t\right)\right)+{\langle \hat{\lambda }\left(t\right),\sum_{k=1}^{p}\sum_{i+j=k-1}C_{k-1}^{i}g_{x^{i}{\left(\dot{x}\right)}^{j}}^{\left(k-1\right)}\left(\bar{x}\left(t\right)\right){\left[h\right]}^{i}{\left(\dot{h}\right)}^{j}\rangle }_{x}\\ \;-\frac{d}{dt}\left[f_{\dot{x}}\left(\bar{x}\left(t\right)\right)+{\langle \hat{\lambda }\left(t\right),\sum_{k=1}^{p}\sum_{i+j=k-1}C_{k-1}^{i}g_{x^{i}{\left(\dot{x}\right)}^{j}}^{\left(k-1\right)}\left(\bar{x}\left(t\right)\right){\left[h\right]}^{i}{\left(\dot{h}\right)}^{j}\rangle }_{\dot{x}}\right]=0,\\ \;\lambda_{i}\left(0\right)=\lambda_{i}\left(2\pi \right),\;\;i=1,2. \end{array}$$

The proof of Theorem 15 is similar to the one for the singular isoperimetric problem in [65].

We now go back to Example 12 for further consideration. The mapping Γ is 2-regular at $\bar{x}\left(t\right)={\left(a\sin t,a\;\cos t,0,0,0\right)}^{T}$ along $h\left(t\right)={\left(a\sin t,a\;\cos t,1,1,1\right)}^{T}$. This means that in this problem p=2.

Consider the following equation$$f_{x}\left(x\right)+{\left(\Gamma^{\prime }\left(x\right)+P_{Y_{2}}\Gamma^{\prime \prime }\left(x\right)h\right)}^{*}\lambda \left(t\right)=0.$$The equation is equivalent to the system of Euler–Lagrange equations(68)$$\left\{\begin{array}{c} 2x_{1}-{\dot{\lambda }}_{1}+\lambda_{2}=0\\ 2x_{2}-{\dot{\lambda }}_{2}-\lambda_{1}=0\\ 2x_{3}+2\lambda_{1}a\sin t+2\lambda_{2}a\cos t=0\\ 2x_{4}+2\lambda_{1}a\cos t-2\lambda_{2}a\sin t=0\\ 2x_{5}+2\lambda_{1}a\left(\cos t-\sin t\right)+2\lambda_{2}a\left(\sin t-\cos t\right)=0.\\ \lambda_{i}\left(0\right)=\lambda_{i}\left(2\pi \right),\;i=1,2. \end{array}\right.$$

One can verify that the following “false solutions” of (Equation 64) and (Equation 65),$$x_{1}=a\sin t,\;x_{2}=a\cos t,\;x_{3}=x_{4}=x_{5}=0,$$
do not satisfy the system (Equation 68) if a≠0. This implies that$$x_{1}=a\sin t,\;x_{2}=a\cos t,\;x_{3}=x_{4}=x_{5}$$
are not solutions to the two-factor Euler–Lagrange Equation (Equation 67) from Theorem 15. Therefore, the only solution to Example 12 is $\bar{x}\left(t\right)={\left(0,0,0,0,0\right)}^{T}.$ Indeed, the two-factor Euler–Lagrange equation in this case has the following form:$$\left\{\begin{array}{c} -{\dot{\lambda }}_{1}+\lambda_{2}=0\\ -{\dot{\lambda }}_{2}-\lambda_{1}=0\\ 2\lambda_{1}a\sin t+2\lambda_{2}a\cos t=0\\ 2\lambda_{1}a\cos t-2\lambda_{2}a\sin t=0\\ 2\lambda_{1}a\left(\cos t-\sin t\right)+2\lambda_{2}a\left(\sin t-\cos t\right)=0.\\ \lambda_{i}\left(0\right)=\lambda_{i}\left(\pi \right),\;i=1,2. \end{array}\right.$$This system has the solution $\bar{x}\left(t\right)={\left(0,0,0,0,0\right)}^{T}$ and ${\bar{\lambda }}_{i}\left(t\right)=0$, i=1,2.

### 4.7. Existence of Solutions to Nonlinear Equations

This section addresses the existence of a solution to an equation of the form (Equation 3), F(x)=0, in the neighbourhood of a chosen point $\bar{x}$. A very general setting is considered, where the function *F* maps from a Banach space *X* to a Banach space *Y*, and the assumptions pertain to the properties of its derivatives in the neighborhood under consideration. This is one of the classical problems of nonlinear analysis, with many important applications, especially in the theory of differential equations (cf. [66,67,68]).

One well-known method for addressing this problem is Newton’s method (see [69]). The solution is obtained as the limit of a recursively defined sequence of approximations. This method is applied in the proof of the first theorem in this section. In particular, the existence of the inverse operator to the derivative of the function at a chosen point is assumed.

The next theorem presented is more general and uses the *p*-factor construction of the operator for functions of class $C^{p+1}$. A certain limitation of this construction is the assumption of the existence of continuous projections onto subspaces of *Y* corresponding to successive orders of the derivatives of the function *F*.

#### 4.7.1. Existence of Solutions to Nonlinear Equations in the Regular Case

Let *X* and *Y* be Banach spaces. Consider a mapping F:X→Y and a problem of existence of a point $\bar{x}$ such that $F(\bar{x})=0.$ We know that equation F(x)=0 is solvable and has a solution $\bar{x}$ when the operator $F^{\prime }\left(x_{0}\right)$ is surjective [27,70]. A modified version of the following theorem was given in [70].

**Theorem** **16.** 
*Let X and Y be Banach spaces, and let $x_{0}\in X$, and let 0<ε<1. Assume $F\in C^{2}\left(B\left(x_{0},\varepsilon \right)\right)$ and $∥F(x_{0})∥=\eta$ for some constant η>0. Suppose that the derivative $F^{\prime }\left(x_{0}\right)$ is invertible and there exist constants δ>0 and C>0 such that $∥{\left(F^{\prime }\left(x_{0}\right)\right)}^{-1}∥=\delta ,$$\underset{x\in B(x_{0},\varepsilon )}{\sup}∥F^{\prime \prime }\left(x\right)∥=C<+\infty$. If, moreover, the following conditions are satisfied:*
 *1.* 

$\;\;\delta \eta \leq \frac{\varepsilon }{2},$

 *2.* 

$\;\;\delta C\varepsilon \leq \frac{1}{4},$

 *3.* $\;\;C\varepsilon \leq \frac{1}{2}$,
*then the equation F(x)=0 has a solution $\bar{x}\in B\left(x_{0},\varepsilon \right).$*


If the first derivative of *F* at $x_{0}$ is not surjective, then Theorem 16 cannot be applied. Consider, for example, a mapping F:R→R defined by$$F\left(x\right)=\frac{1}{7!}x^{7}+x^{5}+\frac{1}{10^{3}}.$$Note that if $x_{0}=0$, the assumptions of Theorem 16 are not satisfied, but the equation F(x)=0 still has a solution $\bar{x}\approx -0.251188$.

#### 4.7.2. Existence of Solutions to Nonlinear Equations in the Singular Case

In this section, we continue considering the problem introduced in Section 4.7.1. Specifically, let *X* and *Y* be Banach spaces, and let F:X→Y. Assume that $F(x_{0})\neq 0$ for some $x_{0}$. We are interested in the existence of a solution $\bar{x}$ to the equation F(x)=0 in some open ball $B(x_{0},\varepsilon )$ of $x_{0}$ such that $F(\bar{x})=0$. Most of the work in solving this problem focuses on Newton’s method or its modifications, under the assumption that $F^{\prime }\left(x_{0}\right)$ is regular (see, e.g., [71]).

Now, consider the degenerate case where $F^{\prime }\left(x_{0}\right)$ is not regular. The focus here is on finding a small constant ε>0 such that the neighborhood $B(x_{0},\varepsilon )$ contains a solution $\bar{x}$ to the equation F(x)=0. We introduce the following notation and assumptions for some p≥2:(69)$$\delta =∥F(x_{0})∥,$$(70)$$\eta =∥{\left\{\Psi_{p}\left(h\right)\right\}}^{-1}∥<\infty ,\;h\in \overset{p}{\underset{k=1}{⋂}}{\mathrm{Ker}}^{k}f_{k}^{\left(k\right)}\left(x_{0}\right),\;∥h∥=1,$$$$c=\max_{k=1,\ldots ,p}\underset{x\in B(x_{0},\varepsilon )}{\sup}∥f_{k}^{\left(k+1\right)}\left(x\right)∥,\;d=4\max_{k=1,\ldots ,p}\frac{1}{(k-1)!}∥f_{k}^{\left(k\right)}\left(x_{0}\right)∥,$$(71)$$\alpha =\min\left\{\frac{3}{4^{p+2}\eta },\min_{k=1,\ldots ,p}\frac{∥f_{k}^{\left(k\right)}\left(x_{0}\right)∥}{(k-1)!}\right\}.$$

The following theorem was proved in [72].

**Theorem** **17.** 
*Let X and Y be Banach spaces, and let F:X→Y be of class $C^{p+1}\left(X\right).$ Assume that there exists $h\in \overset{p}{\underset{k=1}{⋂}}{\mathrm{Ker}}^{k}f_{k}^{\left(k\right)}\left(x_{0}\right),$ with ∥h∥=1, such that F is a p-regular mapping at $\;x_{0}\in X\;$ along h.*

*Assume also that there exists ω,$0<\omega <\frac{1}{2}\nu ,$ where ν∈(0,1), such that the following inequalities hold:*
 *1.* 

$\eta \delta \leq \alpha \frac{\omega^{p}}{2pd},$

 *2.* 

$\frac{4^{p+2}}{3}c\omega \eta \leq \frac{1}{2}.$



*Then the equation F(x)=0 has a solution $\bar{x}=x_{0}+\omega h+\bar{x}\left(\omega \right)\in B_{\nu }\left(x_{0}\right)$, where $\bar{x}\left(\omega \right)$ is a fixed point such that $∥\bar{x}\left(\omega \right)∥\leq \frac{1}{2}\omega .$*


Recall that if our focus is on finding a radius ε>0 such that the open ball $B(x_{0},\varepsilon )$ contains a solution $\bar{x}$ of F(x)=0, then Theorem 17 implies that $\varepsilon =\omega +\bar{x}\left(\omega \right)$. For example, we can take $\varepsilon =\frac{3}{2}\omega .$

As an example of singular nonlinear equation, we consider the problem of existence of local nontrivial solutions of the Boundary Value Problem (BVP) for the ordinary differential equation(72)$$y^{\prime \prime }\left(t\right)+y\left(t\right)+g\left(y\left(t\right)\right)=x\left(t\right)$$
with the boundary conditions(73)y(0)=y(π)=0,
which is degenerate at $\bar{y}\left(t\right)=0$. Here, $y\left(t\right)\in C^{2}\left(\left[0,\pi \right]\right)$ and g,x are given functions such that$$x\in C\left[0,\pi \right],\;g\in C^{p+1}\left(\left[0,\pi \right]\right),\;g\left(0\right)=g^{\prime }\left(0\right)=0.$$

**Remark** **4.** *Recall that the operator $\Psi_{p}$ is defined in *(Equation 19)*. The surjectivity of the operator $\Psi_{p}\left(\omega h\right)$ for any ω≠0 implies the p-regularity condition of the mapping F at the point $x_{0}$ (by the definition). It is also equivalent to the following inequality with a vector h such that ∥h∥=1:*$$∥{\left\{\Psi_{p}\left(\omega h\right)\right\}}^{-1}y∥\leq \left(1+\frac{1}{\omega }+\frac{1}{w^{2}}+\ldots +\frac{1}{\omega^{p-1}}\right).$$

### 4.8. Differential Equations

#### 4.8.1. Nonlinear Boundary-Value Problem

The nonlinear BVP analyzed in this section has the form(74)$$y^{\prime \prime }\left(t\right)+y\left(t\right)+g\left(y\left(t\right)\right)=x\left(t\right)$$
with boundary conditions(75)y(0)=y(π)=0,
where $y\left(t\right)\in C^{2}\left[0,\pi \right]$, x(t)∈C[0,π], and *g* is a $C^{3}$ function from R to R, satisfying(76)$$g\left(0\right)=g^{\prime }\left(0\right)=0,\;x\left(0\right)=x\left(\pi \right)=0.$$

We are interested in the problem of the existence of a solution y(t) to the BVP (Equation 74) and (Equation 75) for given functions x(t) and g(t).

Introduce the notation(77)$$F\left(x,y\right)=y^{\prime \prime }+y+g\left(y\right)-x,$$
and regard *F* as a mapping F:X×Y→Z, where$$X=\left\{x\in C\left[0,\pi \right]|x\left(0\right)=x\left(\pi \right)=0\right\},\;Y=\left\{y\in C^{2}\left[0,\pi \right]|y\left(0\right)=y\left(\pi \right)=0\right\},$$
and Z=C[0,π]. We can rewrite Equation (Equation 74) as(78)F(x,y)=0.The assumptions (Equation 75) and (Equation 76) imply that (0,0) is a solution of (Equation 78): F(0,0)=0. Without loss of generality, we restrict our attention to a neighborhood U×V⊂X×Y of the point (0,0). The problem of existence of a solution y(t) to the BVP (Equation 74) and (Equation 75) for a given function x(t)∈U is equivalent to the problem of existence of an implicit function φ(x):U→Y, such that y=φ(x) and(79)$$F\left(x,y\right)=y^{\prime \prime }+y+g\left(y\right)-x=0,\;y\left(0\right)=y\left(\pi \right)=0.$$

If F(0,0)=0 and the mapping *F* is regular at (0,0)—that is, if the partial derivative of *F* with respect to *y*, denoted $F_{y}^{\prime }\left(0,0\right)$, is a surjective linear operator—then the classical IFT 9 guarantees the existence of a smooth mapping φ defined on a neighborhood of $\bar{x}=0$ such that F(x,φ(x))=0 and φ(0)=0. In this case, the operator $F_{y}^{\prime }\left(0,0\right)$ is given by(80)$$F_{y}^{\prime }\left(0,0\right)y=y^{\prime \prime }+y+g^{\prime }\left(0\right)=y^{\prime \prime }+y,$$
since $g^{\prime }\left(0\right)=0$.

However, the situation changes in the nonregular case. Consider, for example, the BVP$$y^{\prime \prime }\left(t\right)+y\left(t\right)=\sin t,\;y\left(0\right)=y\left(\pi \right)=0,$$
which has no solution. To see this, multiply both sides of the equation by sint and integrate from 0 to π. The left-hand side, after integration by parts, evaluates to zero, while the right-hand side is nonzero. In this example, the operator $F_{y}^{\prime }\left(0,0\right)$ is not surjective, and, therefore, the classical Implicit Function Theorem does not apply to guarantee the existence of a solution to Equation (Equation 78).

#### 4.8.2. Nonlinear Boundary-Value Problem in the Nonregular Case

We consider the boundary-value problem (Equation 74) and (Equation 75) in the nonregular case, using the definitions and notation introduced in Section 4.8.1. Our analysis is restricted to a neighborhood of the point $(\bar{x}\left(t\right),\bar{y}\left(t\right))=\left(0,0\right)$, t∈[0,π].

As shown in Section 4.8.1, the operator $F_{y}^{\prime }\left(0,0\right)$ is not surjective. In this case, we apply the *p*th-order Implicit Function Theorem 11 with p=2 to derive conditions for the existence of an implicit function y=φ(x), and, consequently, for the existence of a solution to the BVP (Equation 74) and (Equation 75).

To apply Theorem 11, we first introduce some auxiliary spaces and functions for the mapping F(x,y), in accordance with Section 4.2.2.

By the definition of the operator $F_{y}^{\prime }\left(0,0\right)$ in (Equation 80), its image is the set of all z(t)∈Z, such that there exists y∈Y satisfying(81)$$y^{\prime \prime }+y=z\left(t\right).$$The general solution of (Equation 81) has the form:$$y\left(t\right)=C_{1}\cos t+C_{2}\sin t-\sin t\int_{0}^{t}\cos\tau z\left(\tau \right)d\tau +\cos t\int_{0}^{t}\sin\tau z\left(\tau \right)d\tau ,\;C_{1},C_{2}\in R.$$Substituting the boundary conditions y(0)=y(π)=0 yields $C_{1}=0$ and$$\int_{0}^{\pi }\sin\tau z\left(\tau \right)d\tau =0.$$Hence,(82)Z1=ImFy′(0,0)=z(t)∈Z∫0πsinτz(τ)dτ=0,
and, as expected, $Z_{1}\neq Z$. The kernel of $F_{y}^{\prime }\left(0,0\right)$ is defined by the boundary value problem$$y^{\prime \prime }+y=0,\;y\left(0\right)=y\left(\pi \right)=0,$$
whose solution is y(t)=Csint, with C∈R. Therefore, $\mathrm{Ker}(F_{y}^{\prime }\left(0,0\right))=\mathrm{span}\left(\sin t\right)$.

Let $Z_{2}$ be a closed complementary subspace to $Z_{1}$. Then, $Z_{2}=\mathrm{span}\left(\sin t\right),$ and the projection operator $P_{Z_{2}}$ is defined as(83)$$P_{Z_{2}}z\left(t\right)=\frac{2}{\pi }\sin t\int_{0}^{\pi }\sin\left(\tau \right)z\left(\tau \right)d\tau ,\;z\left(t\right)\in Z.$$

Next, define the mappings $f_{1}\left(x,y\right)$ and $f_{2}\left(x,y\right)$ by(84)$$f_{1}\left(x,y\right)=F\left(x,y\right),\;f_{2}\left(x,y\right)=P_{Z_{2}}F\left(x,y\right).$$

For p=2, the 2-factor-operator has the form:(85)$$\Psi_{2}\left(h\right)y\left(t\right)={\left(y(t)\right)}^{\prime \prime }+\left(y\left(t\right)\right)+P_{Z_{2}}g^{\prime \prime }\left(0\right)\left[h\right],$$
where h=h(x(t)) is a function.

**Example** **13.** 
*Consider the following nonlinear BVP:*

(86)
$$y^{\prime \prime }\left(t\right)+y\left(t\right)+y^{2}\left(t\right)=v\sin t,\;y\left(0\right)=y\left(\pi \right)=0,$$

*where $g\left(y\right)=y^{2}$, x(t)=vsint, $F\left(x,y\right)=y^{\prime \prime }+y+y^{2}-v\sin t$, v is a constant, and F:X×Y→Z, with X, Y and Z defined above.*
*We now verify that all conditions of the pth-order Implicit Function Theorem 11 are satisfied for the mapping F(x,y) with a sufficiently small v>0 and p=2. Note that $\bar{y}\left(t\right)=0$ is a solution of the homogeneous BVP corresponding to *(Equation 86)*, so that $F(\bar{x},\bar{y})=0$.**For p=2, Condition 1 of Theorem 11 holds for F due to the structure of the mapping g(y), as well as $f_{1}\left(x,y\right)$ and $f_{2}\left(x,y\right)$ introduced in *(Equation 84)*.*
*Condition 2 (the 2-factor-approximation) depends only on the properties of the mapping $g\left(y\right)=y^{2}$ and reduces to the existence of a sufficiently small ε>0 and a neighborhood $U(\bar{y})$ of $\bar{y}$ such that for all $y_{1},y_{2}\in U\left(\bar{y}\right)$,*

$$∥P_{Z_{1}}\left(y_{1}^{2}-y_{2}^{2}\right)∥\leq ∥y_{1}^{2}-y_{2}^{2}∥\leq \varepsilon ∥y_{1}-y_{2}∥,$$

*and*

$$∥P_{Z_{2}}\left(y_{1}^{2}-y_{2}^{2}-y_{1}^{2}+y_{2}^{2}\right)∥\leq \varepsilon \left(∥y_{1}∥+∥y_{2}∥\right)∥y_{1}-y_{2}∥.$$

*Both inequalities hold, and hence Condition 2 is satisfied.*
*Condition 3 is equivalent to the existence of a neighborhood $U(\bar{x})$ such that for some $x\in U(\bar{x})$, there exists a function h=h(x(t)) and $c_{1}>0$ such that*(87)$$h^{\prime \prime }+h+\frac{2}{\pi }\sin t\int_{0}^{\pi }\sin\left(\tau \right)h^{2}\left(\tau \right)d\tau =v\sin t.$$*Problem *(Equation 87)* has an explicit solution*(88)$$h\left(t\right)=\sqrt{\frac{3\pi v}{8}}\sin t,$$*which exists only for v>0. Then, Condition 3 reduces to verifying that there exists a constant $c_{1}>0$ such that*$$∥\sqrt{\frac{3\pi v}{8}}\sin t∥\leq c_{1}{∥v\sin t∥}^{1/2}.$$*This inequality is equivalent to*$$\sqrt{\frac{3\pi }{8}}\leq c_{1},$$*which is satisfied, for instance, by taking $c_{1}=\sqrt{\frac{3\pi }{8}}$.**To verify Condition 4, we observe that with x=vsint and h given by *(Equation 88)*, the set $\Phi_{2}^{-1}\left(-F\left(x,\bar{y}\right)\right)$ consists of a single element {h}. The operator $\Psi_{2}\left(\bar{h}\right)$, defined in *(Equation 85)*, with $\bar{h}\left(t\right)=\sin t,$ takes the form:*$$\Psi_{2}\left(\bar{h}\right)y=y^{\prime \prime }+y+\frac{2}{\pi }\sin t\int_{0}^{\pi }\left(2y\right)\sin^{2}\tau d\tau ,\;$$*which is surjective, and therefore Condition 4 is satisfied.*
*Having verified all four conditions of Theorem 11, we conclude that there exists a solution y(t) to the BVP (Equation 86), satisfying*

$$∥y\left(t\right)∥\leq {c∥v\sin t∥}^{1/2}\leq cv^{1/2},\;c>0.$$



### 4.9. Interpolation by Polynomials

In this section, we consider one of the newest applications of the *p*-regularity theory. There are many books on numerical analysis and numerical methods where the topics of interpolation and polynomial approximation are described in detail (see, for example, [73,74]).

#### 4.9.1. Newton Interpolation Polynomial

Let *f* be $C^{p+1}\left(\left[a,b\right]\right)$ and consider the equationf(x)=0,
where x∈[a,b]. For some Δx>0, define the points $x_{i}$, i=0,…,n, as follows:$$x_{0}=a,\;x_{1}=x_{0}+\Delta x,\;x_{2}=x_{1}+\Delta x,\ldots ,x_{n}=b.$$Let$$y_{i}=f\left(x_{i}\right),\;i=0,1,\ldots ,n.$$

The problem of interpolation is to find a polynomial $P_{n}\left(x\right)$ of degree at most *n* such that $P_{n}\left(x_{i}\right)=y_{i}$, i=0,…,n, and that gives a good approximation of the function f(x).

Let ε=Δx be sufficiently small and assume that $|f\left(x\right)-P_{n}\left(x\right)|\leq C_{1}\varepsilon$, where $C_{1}\geq 0$ is a constant. Assume that the equation f(x)=0 has a solution $\bar{x}\in \left(a,b\right)$, and the equation $P_{n}\left(x\right)=0$ has a solution $\tilde{x}\in \left(a,b\right)$. Our goal is to use the interpolation polynomial $P_{n}\left(x\right)$ and its solution $\tilde{x}$ to obtain the $\varepsilon^{2}$-accuracy of the solution $\bar{x}$, in the sense that(89)$$|\bar{x}-\tilde{x}|\leq C\varepsilon^{2},$$
where C≥0 is a constant. In the regular case, this can be obtained by using, for example, the Newton interpolation polynomial $P_{n}\left(x\right)$ with Δx=ε.

Recall that the Newton interpolation polynomial of degree *n*, related to the data points$$\left(x_{0},y_{0}\right),\left(x_{1},y_{1}\right),\ldots ,\left(x_{n},y_{n}\right),$$
is defined by$$\begin{array}{ll} P_{n}\left(x\right)= & \alpha_{0}+\alpha_{1}\left(x-x_{0}\right)+\alpha_{2}\left(x-x_{0}\right)\left(x-x_{1}\right)+\\ & \ldots +\alpha_{n}\left(x-x_{0}\right)\left(x-x_{1}\right)\ldots \left(x-x_{n-1}\right)\\ & =\sum_{k=0}^{n}\alpha_{k}\omega_{k}\left(x\right), \end{array}$$
where(90)$$\begin{array}{c} \omega_{0}\left(x\right)=1,\\ \omega_{i}\left(x\right)=\left(x-x_{0}\right)\left(x-x_{1}\right)\ldots \left(x-x_{i-1}\right)=\prod_{j=0}^{i-1}\left(x-x_{j}\right),\;i=1,\ldots ,n. \end{array}$$

The coefficients $\alpha_{k}$ are called *divided differences* and are defined using the following relations:(91)$$\alpha_{k}=\left[y_{0},\ldots ,y_{k}\right],\;k=0,1,\ldots ,n,$$
where$$\begin{array}{ccc} \left[y_{k}\right] & & =y_{k},\;k=0,\ldots ,n,\\ \left[y_{k},\ldots ,y_{k+j}\right] & & =\frac{\left[y_{k+1},\ldots ,y_{k+j}\right]-\left[y_{k},\ldots ,y_{k+j-1}\right]}{x_{k+j}-x_{k}}],\\ & & \;k=0,\ldots ,n-j,\;j=1,\ldots ,k. \end{array}$$

In the following example, we consider a nonlinear function f(x), which is not regular at a solution of the equation f(x)=0, and investigate whether a solution of the equation $P_{n}\left(x\right)=0$ provides the desired accuracy (Equation 89) for the solution $\bar{x}$ of f(x)=0, assuming that $|f\left(x\right)-P_{n}\left(x\right)|\leq C_{1}\varepsilon$ holds for a sufficiently small ε.

**Example** **14.** *Consider the function $f\left(x\right)=x^{3}$. The solution of the equation f(x)=0 is $\bar{x}=0$. The function f(x) is singular at $\bar{x}=0$ up to the second order because $f^{\left(i\right)}\left(0\right)=0$ for i=1,2. The goal in this example is to investigate whether the estimate *(Equation 89)* is satisfied when using the interpolation polynomial $P_{1}\left(x\right)$ and a solution of $P_{1}\left(x\right)=0$ to approximate the solution of f(x)=0. Using the equations given above with n=1, we obtain*$$P_{1}\left(x\right)=\alpha_{0}+\alpha_{1}\left(x-x_{0}\right),$$*where the coefficients $\alpha_{0}$ and $\alpha_{1}$ are determined by using Equation *(Equation 91)*.*
*Let ε=Δx be sufficiently small and consider the segment $\left[a,b\right]=\left[-\frac{1}{3}\varepsilon ,\frac{2}{3}\varepsilon \right]$. The interpolation points are $x_{0}=a=-\frac{1}{3}\varepsilon$ and $x_{1}=b=\frac{2}{3}\varepsilon$. Calculating the coefficients, we obtain*

$$\alpha_{0}=f\left(x_{0}\right)=-\frac{\varepsilon^{3}}{27}\;and\;\alpha_{1}=\frac{f\left(x_{1}\right)-f\left(x_{0}\right)}{x_{1}-x_{0}}=\frac{\varepsilon^{2}}{3}.$$

*Hence, the interpolation polynomial has the form*

$$W_{1}\left(x\right)=-\frac{\varepsilon^{3}}{27}+\frac{\varepsilon^{2}}{3}\left(x+\frac{1}{3}\varepsilon \right)=-\frac{\varepsilon^{3}}{27}+\frac{\epsilon^{2}}{3}x+\frac{\varepsilon^{3}}{9}=\frac{2\varepsilon^{3}}{27}+\frac{\epsilon^{2}}{3}x.$$

*Moreover,*

$$|P_{1}\left(x\right)-f\left(x\right)|\approx \varepsilon^{3}\leq C_{2}\varepsilon ,\;C_{2}\geq 0,$$

*for a sufficiently small ε.*
*The solution of the equation $P_{1}\left(x\right)=0$ is*$$\tilde{x}=-\frac{2}{9}\varepsilon ,$$*which is not satisfactory from the approximation accuracy point of view, since*$$|\tilde{x}-\bar{x}|=\left|-\frac{2}{9}\varepsilon -0\right|\approx \varepsilon >\varepsilon^{2}$$*and the desired accuracy *(Equation 89)* is not obtained.*
*Thus, in the degenerate case, contrary to the regular case, while we have the required accuracy of the approximation for the function $f\left(x\right)=x^{3}$, the accuracy of the solution is only of the order ε, and not $\varepsilon^{2}$.*


#### 4.9.2. The *p*-Factor Interpolation Method

In this section, we demonstrate that the desired accuracy (Equation 89) for the solution of the equation f(x)=0 in the degenerate case can be achieved by using the *p*-factor interpolation polynomial, rather than the classical Newton interpolation polynomial, to obtain an approximate solution of f(x)=0.

Let f:R→R be a $C^{p+1}$ function that is singular at a point $\bar{x}$.

For some p>1, we associate *f* with its corresponding *p*-factor function $\bar{f}$, defined as$$\bar{f}\left(x\right)=f\left(x\right)+f^{\prime }\left(x\right)h+\ldots +f^{\left(p-1\right)}\left(x\right){\left[h\right]}^{p-1},$$
where h∈R, h≠0. Similarly to the Newton interpolation method, we construct the *p*-factor interpolation polynomial $\bar{P_{n}}\left(x\right)$ using the function $\bar{f}$ as follows:$$\bar{P_{n}}\left(x\right)=\sum_{k=0}^{n}{\bar{\alpha }}_{k}\omega_{k}\left(x\right),$$
where the functions $\omega_{k}\left(x\right)$ are defined in the same way as in (Equation 90), and the coefficients ${\bar{\alpha }}_{k}$, for k=0,1,…,n, are given by$$\begin{array}{c} {\bar{\alpha }}_{0}=\left[{\bar{y}}_{0}\right]=\bar{f}\left(x_{0}\right),\;\mathrm{and}\;\left[{\bar{y}}_{i}\right]=\bar{f}\left(x_{i}\right),\;\mathrm{for}\;i=1,\ldots ,n,\\ {\bar{\alpha }}_{1}=\left[{\bar{y}}_{0},{\bar{y}}_{1}\right]=\frac{\left[{\bar{y}}_{1}\right]-\left[{\bar{y}}_{0}\right]}{x_{1}-x_{0}},\\ \vdots \\ {\bar{\alpha }}_{n}=\left[{\bar{y}}_{0},\ldots ,{\bar{y}}_{n}\right]=\frac{\left[{\bar{y}}_{1},\ldots ,{\bar{y}}_{n}\right]-\left[{\bar{y}}_{0},\ldots ,{\bar{y}}_{n-1}\right]}{x_{n}-x_{0}}. \end{array}$$

**Theorem** **18.** 
*Let the equation f(x)=0 has a solution $\bar{x}\in \left(a,b\right)$. Assume that $f\in C^{p}\left(\left[a,b\right]\right)$ is p-regular along h≠0 at the point $\bar{x}$. Suppose that $\bar{P_{n}}\left(x\right)$ is the Newton interpolation polynomial for the associated function $\bar{f}$, constructed with a sufficiently small interpolation step ε=Δ>0.*

*Then, the equation $\bar{P_{n}}\left(x\right)=0$ has a solution $\hat{x}\in \left(a,b\right)$ such that*

$$|\hat{x}-\bar{x}|\;\leq \;c\varepsilon^{2},$$

*where c>0 is an independent constant.*


We omit the proof, as it is similar to the proof of convergence of the classical iterative Newton method.

As in the previous sections, we say that a function $f\in C^{p}\left(\left[a,b\right]\right)$ is *p*-regular along h≠0 at the point $\bar{x}\in \left(a,b\right)$ if there is a natural number p≥2 such that$$f^{\left(i\right)}\left(\bar{x}\right)=0,\;i=1,\ldots ,p-1,\;and\;f^{\left(p\right)}\left(\bar{x}\right)\neq 0.$$Note that if p=1, the definition of a *p*-regular function *f* reduces to the standard definition of a regular function, and the *p*-factor interpolation polynomial $\bar{P_{n}}\left(x\right)$ coincides with the classical Newton interpolation polynomial $P_{n}\left(x\right)$.

**Example** **15.** 
*We will apply the p-factor interpolation method to the function from Example 14. Define the function $\bar{f}$ for p=2 and h=1 as*

$$\bar{f}\left(x\right)=f\left(x\right)+f^{\prime }\left(x\right)h+f^{\prime \prime }\left(x\right){\left[h\right]}^{2}=x^{3}+3x^{2}+6x,$$

*and consider the p-factor interpolation polynomial $\bar{P_{1}}\left(x\right)$. Using the same interval as in Example 14, the interpolation points are $x_{0}=a=-\frac{1}{3}\varepsilon$ and $x_{1}=b=\frac{2}{3}\varepsilon$. The coefficients are given by*

$${\bar{\alpha }}_{0}=\bar{f}\left(x_{0}\right)=-\frac{1}{27}\varepsilon^{3}+\frac{1}{3}\varepsilon^{2}-2\varepsilon$$

*and*

$${\bar{\alpha }}_{1}=\frac{\bar{f}\left(x_{1}\right)-\bar{f}\left(x_{0}\right)}{x_{1}-x_{0}}=\frac{1}{3}\varepsilon^{2}+\varepsilon +6.$$

*Thus, the p-factor interpolation polynomial is*

$$\begin{array}{ccc} \bar{P_{1}}\left(x\right) & = & {\bar{\alpha }}_{0}+{\bar{\alpha }}_{1}\left(x-x_{0}\right)\\ & = & -\frac{1}{27}\varepsilon^{3}+\frac{1}{3}\varepsilon^{2}-2\varepsilon +\left(\frac{1}{3}\varepsilon^{2}+\varepsilon +6\right)\left(x+\frac{1}{3}\varepsilon \right)\\ & = & \left(\frac{1}{3}\varepsilon^{2}+\varepsilon +6\right)x+\frac{2}{27}\varepsilon^{3}+\frac{2}{3}\varepsilon^{2}. \end{array}$$

*Hence, for a sufficiently small ε, we have*

$$\left|\bar{P_{1}}\left(x\right)-f\left(x\right)\right|\leq C_{3}\varepsilon ,\;C_{3}\geq 0.$$

*Solving the equation ${\bar{W}}_{1}\left(x\right)=0$, we obtain*$$\hat{x}=-\frac{\frac{2}{27}\varepsilon^{3}+\frac{2}{3}\varepsilon^{2}}{\frac{1}{3}\varepsilon^{2}+\varepsilon +6}=-\frac{\frac{2}{9}\varepsilon^{3}+2\varepsilon^{2}}{\varepsilon^{2}+3\varepsilon +18}.$$*Therefore,*$$|\hat{x}-\bar{x}|=\left|-\frac{\frac{2}{9}\varepsilon^{3}+2\varepsilon^{2}}{\varepsilon^{2}+3\varepsilon +18}-0\right|<\frac{3\varepsilon^{2}}{18}=\frac{\varepsilon^{2}}{6},$$*an we thus obtain the desired $\varepsilon^{2}$-accuracy stated in estimate *(Equation 89)* for the solution of the equation f(x)=0.*

Let us now compare the use of the classical polynomial $P_{1}\left(x\right)$ with the *p*-factor interpolation polynomial $\bar{P_{1}}\left(x\right)$ in approximating the solution $\bar{x}$ of the equation f(x)=0, for the function *f* from Example 14. As mentioned earlier, the polynomial $P_{1}\left(x\right)$ is a good approximation for the function f(x) in the sense that$$|P_{1}\left(x\right)-f\left(x\right)|\approx \varepsilon^{3}\leq C_{2}\varepsilon ,\;C_{2}\geq 0.$$However, the solution $\tilde{x}$ of $P_{1}\left(x\right)=0$ does not yield the desired accuracy for $\bar{x}$, since$$\left|\tilde{x}-\bar{x}\right|\approx \varepsilon \geq C_{4}\varepsilon ,\;C_{4}\geq 0,$$
and the target accuracy of order $\varepsilon^{2}$ is not achieved.

In contrast, the *p*-factor interpolation polynomial $\bar{P_{1}}\left(x\right)$ approximates f(x) with order ϵ:$$\left|\bar{P_{1}}\left(x\right)-f\left(x\right)\right|\approx \varepsilon .$$Thus, using the *p*-factor interpolation polynomial $\bar{P_{1}}\left(x\right)$, we achieve the desired accuracy for the solution $\bar{x}$ of f(x)=0. Specifically, as shown above, the solution $\hat{x}$ of $\bar{P_{1}}\left(x\right)=0$ satisfies estimate (Equation 89):$$|\hat{x}-\bar{x}|\leq \frac{1}{6}\varepsilon^{2}.$$This level of accuracy could not be achieved using the classical interpolation polynomial $P_{1}\left(x\right)$.

## 5. Conclusions

In this paper, we described various applications of the theory of *p*-regularity, including the generalization of the Lyusternik and Implicit Function theorems, the Newton method, optimality conditions for equality and inequality constraints, calculus of variations, and the solvability of nonlinear equations.

We should note that we did not cover all areas where the results of the theory can be applied. In addition, there are other areas of mathematics where the theory of *p*-regularity (or *p*-factor-analysis) has not yet been applied. For example, we did not provide examples of applying the theory to the analysis of the existence of solutions for singular nonlinear partial differential equations, such as the Burger’s nonlinear equation, the Laplace nonlinear differential equation, and others. We also did not cover results related to the existence of solutions depending on a parameter for the Van der Pol differential equation, the Duffing equation, and others. Other results not covered in this paper include examples of applying the theory of *p*-regularity for the analysis of nonlinear dynamical systems, optimality conditions for optimal control problems in the nonregular (degenerate) case. Based on the theory of *p*-regularity, we can develop the theory of so-called *p*-convexity, which can be effective for the analysis of nonlinear problems. Additional information can be found in other studies by the authors.

## Data Availability

The original contributions presented in this study are included in the article. Further inquiries can be directed to the corresponding author(s).

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
