# Peer review of "Towards Nonlinearity: The *p*-Regularity Theory"

_entropy, 2025, doi:10.3390/e27050518_

Round 1

Reviewer 1 Report

Comments and Suggestions for Authors

This paper opens with a review of p-regularity theory, stressing history, application areas, and alternative statements of equivalent conditions. It goes on to focus on specific applications, giving several examples. Much of this material can be found in more concise forms in the existing literature; however, as a reasonably self-contained overview, this paper could be a valuable resource to one new to the field. In general, it is clearly written, with an exceptional range of examples. Therefore I recommend it be accepted for publication after revisions to address the following points.

The abstract is rather long, and a significant part of it is devoted to listing potential application areas without specifying its advantages to them. The abstract would be more useful in describing the mathematical content of the paper if this material were reduced or eliminated.

The Conclusion reads like a collection of results that didn't fit elsewhere, not a summary of the foregoing work. It does not do justice to the rest of the paper, and needs substantial revision.

Several of the references are works by one or more of the current authors. Without meaning to question the relevance of any of these, I must ask if it is necessary to cite them all, especially the earlier publications.

The heading "Introduction" is repeated (on lines 26 and 27), resulting in section 1 being empty and affecting the numbering of the following sections.

Heading 5.5, "Modified Lagrange function method", is immediately followed by heading 5.5.1 with identical text. One would expect the lower-level heading to be more restrictive or otherwise different from its parent.

"Van der Pol" is misspelled in line 960.

Author Response

Thank you for your time and valuable comments on our article. We have provided responses to the raised issues, which can be found in the attached document.

Reviewer 2 Report

Comments and Suggestions for Authors

Dear Authors,

your valuable work looks as if it had multiple authors. That's what it is. Therefore, you must compose a manuscript with unique item labels. I noted several formulas that are interrupted from the sky to the ribs, as we say in Croatia.

In counting the $p$-regularity applications, you lack references. The manuscript is further aggravated by the addition of formulas and clumsy Roman numerals writing. There are theorems without proper preambles. I notice all in the comments below. Several incomplete definitions are connoted. They are connoted in the sense that a reader can assume properly what the Authors mean. We can't presume it. Don't you agree? I think that a reader does not have to pass any exam in front of you if he wants to read your article. 

I recommend you to improve your manuscript to be available for a wide circle of readers who will be interested in Entropy journal. 

All my comments are attached below. 

The manuscript is worth to be published because it gives a view on widely appliable p-regularity theory.

Author Response

(The authors gave the same response as above.)

Round 2

Reviewer 1 Report

Comments and Suggestions for Authors

The authors have adequately addressed the concerns raised in my review of the original version of this manuscript, and I am pleased to recommend that the revised paper be accepted for publication.

Author Response

Thank you for reviewing our article. We appreciate your time and effort spent in reviewing our manuscript.

Reviewer 2 Report

Comments and Suggestions for Authors

Dear Authors, 

Your improvements make the reading much easier. The second reading is much easier itself, and I can't get into the first-reader skin. But I find a few places where you can make reading easier for the first readers. 

Now it is easier to find some doubts about the terms' definitions in a mathematical sense. In the file attached below I insert all my comments and suggestions. Additional file I will send by the Editors.

Maybe some comments are the consequences of my misunderstanding. But the manuscript must be clear for reading. Especially for those readers who could apply something from your p-regularity theory review.

Your manuscript still looks like several authors created it without meeting themselves.

Comments on the Quality of English Language

All my comments are included in the attached file. Since I am not an English language expert, I only suggest that the authors consult a good English lecturer friend.

Author Response

(The authors gave the same response as above.)

Round 3

Reviewer 2 Report

Comments and Suggestions for Authors

Dear Authors, 

In any reading, mistakes appear. It is a fact. At first, I thought that only cosmetic comments I would have.  Until I found subsection 4.6.1. In this subsection I found many inconsistencies that must be improved. 

I am very sorry that I didn't find them during the last revision, but I need to see your manuscript once more. 

Please, check all once more although is impossible to find own mistake. I hope next time I will finish with revisions. Sincerely, your reviewer. 

Author Response

We would like to sincerely thank the reviewer for all the valuable comments and suggestions. We have carefully addressed all of them in the revised version of the manuscript.  Attached is our reply letter to the reviewer.

Round 4

Reviewer 2 Report

Comments and Suggestions for Authors

Dear Authors, 

You made a very good manuscript putting many p-regularity applications in several different mathematical branches. During these revisions, you did almost everything to make the manuscript suitable for readers possibly interested in p-regulatority.

Unfortunately, in the last revision, I noticed a crucial mistake in a mathematical sense. And now I am not satisfied with how you improved the formulas. 

Again, I try to explain my comments and requests more clearly. Besides the file attached, I will ask the Editors to send you one more document.   

Comments on the Quality of English Language

The language is understandable. I only noticed some phrases I commented in a manner to sound better.